# Understanding the Under-Coverage Bias in Uncertainty Estimation

**Yu Bai**
Salesforce Research
yu.bai@salesforce.com

**Song Mei**
UC Berkeley
songmei@berkeley.edu

**Huan Wang**
Salesforce Research
huan.wang@salesforce.com

**Caiming Xiong**
Salesforce Research
cxiong@salesforce.com

## Abstract

Estimating the data uncertainty in regression tasks is often done by learning a quantile function or a prediction interval of the true label conditioned on the input. It is frequently observed that quantile regression—a vanilla algorithm for learning quantiles with asymptotic guarantees—tends to *under-cover* than the desired coverage level in reality. While various fixes have been proposed, a more fundamental understanding of why this under-coverage bias happens in the first place remains elusive.

In this paper, we present a rigorous theoretical study on the coverage of uncertainty estimation algorithms in learning quantiles. We prove that quantile regression suffers from an inherent under-coverage bias, in a vanilla setting where we learn a realizable linear quantile function and there is more data than parameters. More quantitatively, for $\alpha > 0.5$ and small $d/n$, the $\alpha$-quantile learned by quantile regression roughly achieves coverage $\alpha - (\alpha - 1/2) \cdot d/n$ regardless of the noise distribution, where $d$ is the input dimension and $n$ is the number of training data. Our theory reveals that this under-coverage bias stems from a certain high-dimensional parameter estimation error that is not implied by existing theories on quantile regression. Experiments on simulated and real data verify our theory and further illustrate the effect of various factors such as sample size and model capacity on the under-coverage bias in more practical setups.

## 1 Introduction

This paper is concerned with the problem of uncertainty estimation in regression problems. Uncertainty estimation is an increasingly important task in modern machine learning applications—Models should not only make high-accuracy predictions, but also have a sense of how much the true label may deviate from the prediction. This capability is crucial for deploying machine learning in the real world, in particular in risk-sensitive domains such as medical AI [15, 29], self-driving cars [47], and so on. A common approach for uncertainty estimation in regression is to learn a *quantile function* or a *prediction interval* of the true label conditioned on the input, which provides useful distributional information about the label. Such learned quantiles are typically evaluated by their *coverage*, i.e., probability that it covers the true label on a new test example. For example, a learned 90% upper quantile function should be an actual upper bound of the true label at least 90% of the time.

Algorithms for learning quantiles date back to the classical quantile regression [35], which estimates the quantile function by solving an empirical risk minimization problem with a suitable loss function that depends on the desired quantile level $\alpha$. Quantile regression is conceptually simple, and is

35th Conference on Neural Information Processing Systems (NeurIPS 2021).

theoretically shown to achieve asymptotically correct coverage as the sample size goes to infinity [34] or approximately correct coverage in finite samples under specific modeling assumptions [46, 60, 56]. However, it is observed that quantile regression often *under-covers* than the desired coverage level in practice [53]. Various alternative approaches for constructing quantiles and confidence intervals are proposed in more recent work, for example by aggregating multiple predictions using Bayesian neural networks or ensembles [24, 37], or by building on the conformal prediction technique to construct prediction intervals with finite-sample coverage guarantees [68, 66, 39, 53]. However, despite these advances, a more fundamental understanding on why vanilla quantile regression exhibits this under-coverage bias is still lacking.

This paper revisits quantile regression and presents a first precise theoretical study on its coverage, in a new regime where the number of samples $n$ is proportional to the dimension $d$, and the ratio $d/n$ is small (so that the problem is under-parametrized). Our main result shows that quantile regression exhibits an inherent under-cover bias under this regime, even in the well-specified setting of learning a linear quantile function when the true data distribution follows a Gaussian linear model. To the best of our knowledge, this is the first rigorous theoretical justification of the under-coverage bias. Our main contributions are summarized as follows.

- *We prove that linear quantile regression exhibits an inherent under-coverage bias* in the well-specified setting where the data is generated from a Gaussian linear model, and the number of samples $n$ is proportional to the feature dimension $d$ with a small $d/n$ (Section 3). More quantitatively, quantile regression at nominal level $\alpha \in (0.5, 1)$ roughly achieves coverage $\alpha - (\alpha - 1/2)d/n$ regardless of the noise distribution. To the best of our knowledge, this is the first rigorous characterization of the under-coverage bias in quantile regression.

- Towards understanding the source of this under-coverage bias, we disentangle the effect of estimating the bias and estimating the linear coefficient on the coverage of the learned linear quantile (Section 4). We show that the estimation error in the bias can have either an under-coverage or over-coverage effect, depending on the noise distribution. In contrast, the estimation error in the linear coefficient always drives the learned quantile to under-cover, and we show this effect is present even on broader classes of data distributions beyond the Gaussian linear model.

- We perform experiments on simulated and real data to test our theory (Section 5). Our simulations show that the coverage of quantile regression in Gaussian linear models agrees well with our precise theoretical formula as well as the $\alpha - (\alpha - 1/2)d/n$ approximation. On real data, we find quantile regression using high-capacity models (such as neural networks) exhibits severe under-coverage biases, while linear quantile regression can also have a mild but non-negligible amount of under-coverage, even after we remove the potential effect of model misspecification.

- On the technical end, our analysis builds on recent understandings of empirical risk minimization problems in the high-dimensional proportional limit with a small $d/n$, and develops new techniques such as a novel concentration argument to deal with an additional learnable variable in learning linear models with biases, which we believe could be of further interest (Section 6).

## 1.1 Related work

**Algorithms for uncertainty estimation in regression** The earliest methods for uncertainty estimation in regression adopted subsampling methods (bootstrap) or leave-one-out methods (Jackknife) for assessing or calibrating prediction uncertainty [52, 63, 58, 25]. More recently, a growing line of work builds on the idea of conformal prediction [55] to design uncertainty estimation algorithms for regression. These algorithms provide confidence bounds or prediction intervals by post-processing any predictor, and can achieve distribution-free finite-sample marginal coverage guarantees utilizing exchangeability of the data [50, 66, 39, 53, 33, 67, 69, 68, 11]. Further modifications of the conformal prediction technique can yield stronger guarantees such as group coverage [10] or coverage under distribution shift [9] under additional assumptions. Our under-coverage results advocate the necessity of such post-processing techniques, and are complementary in the sense that we provide understandings on the more vanilla quantile regression algorithm. Quantiles and prediction intervals can also be obtained by aggregating multiple predictors, such as using Bayesian neural networks [41, 24, 32, 44, 42] or ensembles [37, 49, 28, 45]. These methods offer an alternative approach for uncertainty estimation, but do not typically come with coverage guarantees.

**Theoretical analysis of quantile regression**  Linear quantile regression with the pinball loss dates back to the late 1970s [34]. The same work proved the asymptotic normality of the regression coefficients in the $n \to \infty$, fixed $d$ limit. Takeuchi et al. [60] studied non-parametric quantile regression using kernel methods, and provided generalization bounds (with the pinball loss) based on the Rademacher complexity. Meinshausen [46] studied non-parametric quantile regression using random forest and showed its consistency under proper assumptions. Christmann and Steinwart [19], Steinwart et al. [56] established a "self-calibration" inequality for the quantile loss, which, when combined with standard generalization bounds, can be translated to an estimation error bound for quantile regression. These works all focus on bounding the parameter or function estimation error, which can be translated to bounds on the coverage bias, but does not tell the sign of this coverage bias as we do in this paper. We also remark that conformalization can be used in conjunction with quantile regression to correct its coverage bias [53].

**Uncertainty quantification for classification**  For classification problems, two main types of uncertainty quantification methods have been considered: outputting discrete prediction sets with guarantees of covering the true (discrete) label [70, 71, 38, 7, 12, 18, 17], or calibrating the predicted probabilities [51, 72, 73, 37, 26]. The connection between prediction sets and calibration was discussed in [27]. The sample complexity of calibration has been studied in a number of theoretical works [36, 27, 54, 30, 40, 8]. Our work is inspired by the recent work of Bai et al. [8], which showed that logistic regression is over-confident even if the model is correctly specified and the sample size is larger than the dimension.

**High-dimensional behaviors of empirical risk minimization**  There is a rapidly growing literature on limiting characterizations of convex optimization-based estimators in the $n \propto d$ regime [21, 13, 23, 31, 57, 61, 20, 62, 43, 59, 16]. Our analysis builds on results for unregularized M-estimator derived in [62] and generalizes theirs in certain aspects (see also [23, 20, 31]).

## 2  Preliminaries

In this paper we focus on the problem of learning quantiles. Suppose we observe a training dataset $\{(\mathbf{x}_i, y_i)\}_{i=1}^{n}$ drawn i.i.d. from some joint distribution $\mathbb{P}$ on $\mathbb{R}^d \times \mathbb{R}$, where $\mathbf{x}_i \in \mathbb{R}^d$ is the input features and $y \in \mathbb{R}$ is the real-valued response (label). Let $F(t|\mathbf{x}) := \mathbb{P}(Y \leq t|\mathbf{X} = \mathbf{x})$ denote the conditional CDF of $Y|\mathbf{X}$. Our goal is to learn the $\alpha$-(conditional) quantile of $Y|\mathbf{X}$:

$$q_\alpha^\star(\mathbf{x}) := \inf \{t \in \mathbb{R} : F(t|\mathbf{x}) \geq \alpha\}.$$

For example, $q_{0.95}^\star(\mathbf{x})$ is the ground truth 95% quantile of the true conditional distribution $Y|\mathbf{X}$, and can be seen as the "ideal" 95% upper confidence bound for the label $y$ given the features $\mathbf{x}$. Throughout this paper we work with upper quantiles, that is, $\alpha \in (0.5, 1)$ (some typical choices are $\alpha \in \{0.8, 0.9, 0.95\}$); by symmetry our results hold for learning lower quantiles as well.

**Coverage**  For any learned quantile function $\widehat{f} : \mathbb{R}^d \to \mathbb{R}$, the marginal coverage (henceforth "coverage") of $\widehat{f}$ is the probability of $y \leq \widehat{f}(\mathbf{x})$ on a new test example $(\mathbf{x}, y)$:

$$\mathrm{Coverage}(\widehat{f}) := \mathbb{P}_{(\mathbf{x},y)}\Big(y \leq \widehat{f}(\mathbf{x})\Big) = \mathbb{E}_{\mathbf{x}}\Big[\mathbb{P}\Big(y \leq \widehat{f}(\mathbf{x})|\mathbf{x}\Big)\Big]. \tag{1}$$

For learning the $\alpha$-quantile ($\alpha > 0.5$), we usually expect $\mathrm{Coverage}(\widehat{f}) \approx \alpha$, i.e. $\widehat{f}(\mathbf{x})$ covers the label $y$ on approximately $\alpha$ proportion of the data, under the ground truth data distribution.

We say that $\widehat{f}$ has *under-coverage* if $\mathrm{Coverage}(\widehat{f}) < \alpha$ and *over-coverage* if $\mathrm{Coverage}(\widehat{f}) > \alpha$. Note that these two notions are not symmetric: Over-coverage means that the learned upper quantile $\widehat{f}(\mathbf{x})$ is overly conservative (higher than enough), and is typically tolerable; In contrast, under-coverage means that $\widehat{f}(\mathbf{x})$ fails to cover $y$ with $\alpha$ probability, and is typically considered as a failure. We remark that while there exist more fine-grained notions of coverage such as conditional coverage [10], the (marginal) coverage is still a basic requirement for any quantile learning algorithm.

**Quantile regression**  We consider quantile regression, a standard method for learning quantiles from data [35]. Quantile regression estimates the true quantile function $q_\alpha(\cdot)$ via the *pinball loss* [34, 56]

$$\ell^\alpha(t) = -(1-\alpha)t\mathbf{1}\{t \leq 0\} + \alpha t\mathbf{1}\{t > 0\}. \tag{2}$$

Note that in the special case of $\alpha = 0.5$, we have $\ell^{0.5}(t) = |t|/2$, and thus the pinball loss strictly generalizes the absolute loss (for learning medians) to learning any quantile. Given the training dataset and any function class $\{f_\theta : \theta \in \Theta\}$ (e.g. linear models or neural networks), quantile regression solves the (unregularized) empirical risk minimization (ERM) problem

$$\widehat{\theta} = \arg\min_{\theta \in \Theta} \widehat{R}_n(\theta) := \frac{1}{n} \sum_{i=1}^{n} \ell^\alpha(y_i - f_\theta(\mathbf{x}_i)). \tag{3}$$

(We take $\widehat{\theta}$ as any minimizer of $\widehat{R}_n$ when the minimizer is non-unique.) Let $R(\theta) := \mathbb{E}[\widehat{R}_n(\theta)]$ denote the corresponding population risk. It is known that the population risk over all (measurable) functions is minimized at the true quantile $q_\alpha^\star = \arg\min_f R(f)$ under minimal regularity conditions (for completeness, we provide a proof in Appendix B.1).

## 3   Quantile regression exhibits under-coverage

We analyze quantile regression in the vanilla setting where the input distribution is a standard Gaussian and $y$ follows a linear model of $\mathbf{x}$:

$$y = \mathbf{w}_\star^\top \mathbf{x} + z, \quad \text{where} \quad \mathbf{x} \sim \mathsf{N}(\mathbf{0}, \mathbf{I}_d), \;\; z \sim P_z. \tag{4}$$

Above, $\mathbf{w}_\star \in \mathbb{R}^d$ is the ground truth coefficient vector, and the noise $z \sim P_z$ is independent of $\mathbf{x}$. The Gaussian input assumption is required only for technical convenience in the high-dimensional limiting analysis, and we believe it is not strictly required for the same result to hold[1] (an extension to more general input distributions can also be found in Section 4). The noise distribution $P_z$ is required to satisfy the following smoothness assumption, but can otherwise be arbitrary:

**Assumption A** (Smooth density). *The noise distribution $P_z$ has a smooth density $\phi_z \in C^\infty(\mathbb{R})$ (with corresponding CDF $\Phi_z$), with bounded derivatives: $\sup_{t \in \mathbb{R}} |\phi_z^{(k)}(t)| < \infty$ for any $k \geq 0$. We further assume that $\phi_z(z_\alpha) > 0$, where $\alpha \in (0.5, 1)$ is our pre-specified quantile level, and $z_\alpha := \inf\{t \in \mathbb{R} : \Phi_z(t) \geq \alpha\}$ is the $\alpha$-quantile of $P_z$.*

Under the above model, it is straightforward to see that the true $\alpha$-conditional quantile of $y|\mathbf{x}$ is also a linear model (with bias):

$$q_\alpha^\star(\mathbf{x}) = \mathbf{w}_\star^\top \mathbf{x} + z_\alpha. \tag{5}$$

Given the training data $\{(\mathbf{x}_i, y_i)\}_{i=1}^n$, we learn a linear quantile function $\widehat{f}(\mathbf{x}) = \widehat{\mathbf{w}}^\top \mathbf{x} + \widehat{b}$ via quantile regression:

$$(\widehat{\mathbf{w}}, \widehat{b}) = \arg\min_{\mathbf{w}, b} \widehat{R}_n(\mathbf{w}, b) := \frac{1}{n} \sum_{i=1}^{n} \ell^\alpha(y_i - (\mathbf{w}^\top \mathbf{x}_i + b)), \tag{6}$$

where $\ell^\alpha$ is the pinball loss in (2). As our linear function class realizes the true quantile function (5), the population risk is minimized at the true quantile: $\arg\min_{\mathbf{w}, b} R(\mathbf{w}, b) = (\mathbf{w}_\star, z_\alpha)$.

We are now ready to state our main result, which shows that quantile regression exhibits an inherent under-coverage bias even in this vanilla realizable setting.

**Theorem 1** (Quantile regression exhibits under-coverage bias). *Suppose the data is generated from the linear model (4) and the noise satisfies Assumption A. Let $\widehat{f}(\mathbf{x}) = \widehat{\mathbf{w}}^\top \mathbf{x} + \widehat{b}$ be the output of quantile regression (6) at level $\alpha \in (0.5, 1)$. Then, in the limit of $n, d \to \infty$ and $d/n \to \kappa$ where $\kappa \in (0, \kappa_0]$ for some small $\kappa_0 > 0$, for the coverage (1), we have ($\xrightarrow{p}$ denotes convergence in probability)*

$$\text{Coverage}(\widehat{f}) \xrightarrow{p} \alpha - C_{\alpha, \kappa} \quad \text{for some } C_{\alpha, \kappa} > 0.$$

*That is, the limiting coverage of the learned quantile function is less than $\alpha$. Further, for small enough $\kappa$ we have the local linear expansion*

$$C_{\alpha, \kappa} = (\alpha - 1/2)\kappa + o(\kappa). \tag{7}$$

---

[1]Our results can be extended directly to any correlated Gaussian input $\mathbf{x} \sim \mathsf{N}(\mathbf{0}, \mathbf{\Sigma})$ by the transform $\widetilde{\mathbf{x}} = \mathbf{\Sigma}^{-1/2}\mathbf{x}$ and $\widetilde{\mathbf{w}}_\star = \mathbf{\Sigma}^{1/2}\mathbf{w}_\star$. We believe our results also hold for i.i.d. sub-Gaussian inputs by the universality principle (e.g. [14]).

Theorem 1 builds on the precise characterization of ERM problems in the high-dimensional proportional limit [62], along with new techniques over existing work for dealing with the unique challenges in quantile regression (such as analyzing the additional learnable bias $b$). An overview of the main technical steps is provided in Section 6, and the full proof is deferred to Appendix C.

**Implications** Theorem 1 can be illustrated by the following numeric example. Suppose we perform quantile regression at $\alpha = 0.9$, where the data follows the linear model (4), and our $\kappa = d/n = 0.1$ (so that the sample size is 10x number of parameters). Then Theorem 1 shows that, even in this realizable, under-parametrized setting, the coverage of the learned quantile $\widehat{f}$ is going to be roughly $0.9 - C_{\alpha,\kappa}$ when $n, d$ are large, and further $C_{\alpha,\kappa} \approx (\alpha - 1/2)\kappa = 0.04$. Thus the actual coverage is around $0.9 - 0.04 = 0.86$, and such a 4% under-coverage bias can be rather non-negligible in reality.

We remark that a symmetric conclusion of Theorem 1 also holds for lower quantiles, and we further expect similar results also hold for learning prediction intervals, where the coverage is defined as the two-sided coverage of the prediction interval formed by the learned {lower quantile, upper quantile}. To the best of our knowledge, this offers a first precise theoretical understanding of why practically trained quantiles or prediction intervals often under-cover than the desired coverage level [53].

**Comparison against existing theories** An important feature of the under-coverage bias shown in Theorem 1 is that it only shows up in the $n, d$ proportional regime, and is not implied by existing theories on quantile regression. Classical asymptotic theory only shows asymptotic normality $\sqrt{n}([\widehat{\mathbf{w}}, \widehat{b}] - [\mathbf{w}_\star, z_\alpha]) \to \mathsf{N}(\mathbf{0}, \mathbf{V})$ in the $n \to \infty$, fixed $d$ limit [34, 64]. Under this limit, $\mathrm{Coverage}(\widehat{f})$ is consistent at $\alpha$ with $O(1/\sqrt{n})$ deviation. Christmann and Steinwart [19], Steinwart et al. [56] consider the finite $n, d$ setting and establish *self-calibration inequalities* (similar to strong convexity) that bounds the quantile estimation error by the square root excess loss $\sqrt{R(\widehat{f}) - R(q_\alpha^\star)}$. Combined with standard generalization theories (e.g. via Rademacher complexities) and Lipschitzness, this can be turned into a bound on $|\mathrm{Coverage}(\widehat{f}) - \alpha|$, but does not tell the sign (positive or negative) of the coverage bias.

**Large $\kappa$; extension to over-parametrized learning** While Theorem 1 requires a small $\kappa = d/n$, the approximation formula (7) suggests that the under-coverage should get more severe as $\kappa$—the measure of over-parametrization in this problem—gets larger. We confirm this trend experimentally in our simulations in Section 5.1.

As an extension to Theorem 1, we also show theoretically that the under-coverage bias indeed becomes even more severe in over-parametrized learning, under the same linear model (4): When $d/n > \widetilde{O}(1)$, and the noise $P_z$ is sub-Gaussian and symmetrically distributed about 0, the convergence point of the gradient descent path on the quantile regression risk $\widehat{R}_n$ is the minimum-norm interpolator of the data, which has coverage $0.5 \pm \widetilde{O}(1/\sqrt{d})$ with high probability (see Appendix D for the formal statement and the proof). Notably, this 0.5 coverage does not depend on $\alpha$ and exhibits a severe under-coverage.

## 4 Understanding the source of the under-coverage bias

In this section, we take steps towards a deeper understanding of how the under-coverage bias shown in Theorem 1 happens. Recall that the quantile regression returns $\widehat{f}(\mathbf{x}) = \widehat{\mathbf{w}}^\top \mathbf{x} + \widehat{b}$ where $(\widehat{\mathbf{w}}, \widehat{b})$ is a solution to the ERM problem (6) and estimates the true parameters $(\mathbf{w}_\star, z_\alpha)$. Our main approach in this section is to *disentangle the effect of the two sources*—the estimation error in $\widehat{b}$ and the estimation error in $\mathbf{w}$—on the coverage of $\widehat{f}$.

We show that the estimation error in $\widehat{b}$ *can have either an under-coverage or an over-coverage effect*, depending on the noise distribution (Section 4.1). In contrast, the estimation error in $\widehat{\mathbf{w}}$ *always has an under-coverage effect*; this holds not only for the linear model assumed in Theorem 1, but also on more general data distributions (Section 4.2). In the setting of Theorem 1, this under-coverage effect of $\widehat{\mathbf{w}}$ is always strong enough to dominate the effect of $\widehat{b}$, leading to the overall under-coverage.

## 4.1 Effect of estimation error in $\widehat{b}$

To study the effect of $\widehat{b}$, we use the quantity $\widehat{b} - z_\alpha$ as a measure for its effect on the coverage—Recall that the true quantile is $q_\alpha^\star(\mathbf{x}) = \mathbf{w}_\star^\top \mathbf{x} + z_\alpha$, thus having $\widehat{b} < z_\alpha$ means that $\widehat{b}$ contributes to under-coverage, whereas $\widehat{b} > z_\alpha$ means $\widehat{b}$ contributes to over-coverage. (This can be seen more straightforwardly in the easier case where we know $\mathbf{w}_\star$ and only output $\widehat{b}$ to estimate $z_\alpha$.)

The following corollary shows that, under the same settings of Theorem 1, the error $\widehat{b} - z_\alpha$ can be understood precisely. The proof can be found in Appendix E.1.

**Corollary 2** (Effect of $\widehat{b}$ on coverage depends on noise distribution). *Under the same settings as Theorem 1, for any $\alpha \in (0.5, 1)$, as $n, d \to \infty$ with $d/n \to \kappa \in (0, \kappa_0]$, we have*

(a) *The learned bias $\widehat{b}$ from quantile regression* (6) *converges to the following limit:*

$$\widehat{b} - z_\alpha \xrightarrow{p} C_{\alpha,\kappa}^b = \bar{b}_0 \kappa + o(\kappa),$$

*where $\bar{b}_0$ has a closed-form expression:*

$$\bar{b}_0 := \frac{-\alpha(1-\alpha)\phi_z'(z_\alpha) - (2\alpha - 1)\phi_z^2(z_\alpha)}{2\phi_z^3(z_\alpha)}. \tag{8}$$

(b) *For any $\alpha \in (0.5, 1)$, when $P_z$ is the Gaussian distribution (with arbitrary scale), we have $\bar{b}_0 < 0$ in which case $C_{\alpha,\kappa}^b < 0$ for small enough $\kappa$. Conversely, for any $\alpha \in (0.5, 1)$, there exists some noise distribution $P_z$ for which $\bar{b}_0 > 0$, in which case $C_{\alpha,\kappa}^b > 0$ for small enough $\kappa$.*

Corollary 2 shows that the sign of $C_{\alpha,\kappa}^b$ in the limiting regime (and thus the effect of $\widehat{b}$ on the coverage) depends on $\bar{b}_0$, which in turn depends on the noise distribution $P_z$. For common noise distributions such as Gaussian we have $C_{\alpha,\kappa}^b < 0$ at small $\kappa$, but there also exists $P_z$ such that $C_{\alpha,\kappa}^b > 0$. Note that the second claim in part (b) follows directly from (8): we can always design the density $\phi_z$ by varying $\phi_z(z_\alpha)$ and $\phi_z'(z_\alpha)$ so that $\bar{b}_0 > 0$. Overall, this result shows that the under-coverage bias in Theorem 1 cannot be simply explained by the under-estimation error in $\widehat{b}$.

## 4.2 Effect of estimation error in $\widehat{\mathbf{w}}$; relaxed data distributions

We now show that the primary source of the under-coverage is the estimation error in $\widehat{\mathbf{w}}$, which happens not only on the linear data distribution assumed in Theorem 1, but also on a broader class of data distributions. We consider the following relaxed data distribution assumption

$$y = \mu_\star(\mathbf{x}) + \sigma_\star(\mathbf{x})z, \tag{9}$$

where the noise $z \sim P_z$. We do not put structural assumptions on $(\mu_\star, \sigma_\star)$, except that we assume the true $\alpha$-quantile is still a linear function of $\mathbf{x}$, that is, there exists $(\mathbf{w}_\star, b_\star)$ for which

$$q_\alpha^\star(\mathbf{x}) = \mu_\star(\mathbf{x}) + \sigma_\star(\mathbf{x})z_\alpha = \mathbf{w}_\star^\top \mathbf{x} + b_\star. \tag{10}$$

Since here we are interested in the effect of estimating $\mathbf{w}_\star$, for simplicity, we assume that we know $b_\star$ and only estimate $\mathbf{w}_\star$ via some estimator $\widehat{\mathbf{w}}$. We now collect our assumptions and state the result.

**Assumption B** (Relaxed data distribution). *The data is distributed as model* (9) *with a linear $\alpha$-quantile function* (10). *Further, the data distribution satisfies the following regularity conditions:*

(a) *The distribution of $\mathbf{x} \in \mathbb{R}^d$ is symmetric about $\mathbf{0}$, has a lower bounded covariance $\mathbb{E}[\mathbf{x}\mathbf{x}^\top] \succeq \underline{\gamma}\mathbf{I}_d$, and is $K$-sub-Gaussian, for constants $\underline{\gamma}, K > 0$.*

(b) *The variance function $\sigma_\star(\cdot)$ is bounded and symmetric: For all $\mathbf{x} \in \mathbb{R}^d$ we have $\underline{\sigma} \leq \sigma_\star(\mathbf{x}) \leq \overline{\sigma}$ for some constants $\underline{\sigma}, \overline{\sigma} > 0$, and $\sigma_\star(\mathbf{x}) = \sigma_\star(-\mathbf{x})$.*

(c) *The noise density $\phi_z$ is continuously differentiable and symmetric about 0, i.e. $\phi_z(t) = \phi_z(-t)$ for all $t \in \mathbb{R}$. Further, $\phi_z$ is uni-modal, i.e. $\phi_z'(t)|_{t<0} > 0$ and $\phi_z'(t)|_{t>0} < 0$.*

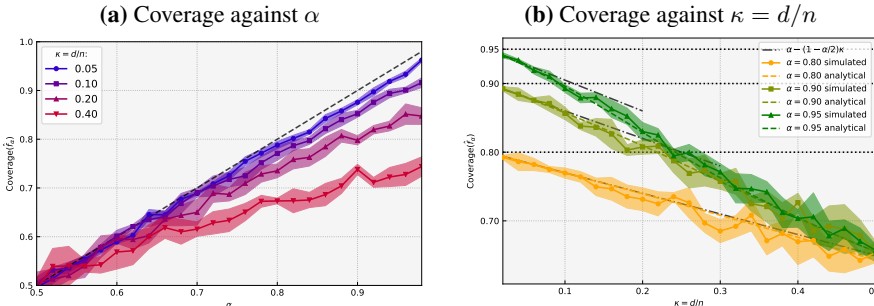

**Figure 1:** Coverage of quantile regression on simulated data from the realizable linear model (4). **(a)(b)** Each dot represents a combination of $(\alpha, \kappa)$ and reports the mean and one-std coverage over 8 random problem instances. **(a)** Coverage against the nominal quantile level $\alpha$ for fixed values of $\kappa = d/n$. **(b)** Coverage against $\kappa$ for fixed $\alpha \in \{0.8, 0.9, 0.95\}$. Here "analytical" refers to our analyitical formula $\alpha - C_{\alpha, \kappa}$, and $\alpha - (1 - \alpha/2)\kappa$ is its local linear approximation at small $\kappa$ (both from Theorem 1).

**Theorem 3** (Estimation error in $\widehat{\mathbf{w}}$ leads to under-coverage on a family of data distributions)**.** *Under the relaxed data distribution assumption (Assumption B), for any $\alpha > 3/4$, there exists constants $c, r_0 > 0$ such that for any learned quantile estimate $\widehat{f}(\mathbf{x}) = \widehat{\mathbf{w}}^\top \mathbf{x} + b_\star$ with small estimation error $\|\widehat{\mathbf{w}} - \mathbf{w}_\star\|_2 \leq r_0$, we have*

$$\text{Coverage}(\widehat{f}) \leq \alpha - c\underline{\gamma}/\overline{\sigma}^2 \cdot \|\widehat{\mathbf{w}} - \mathbf{w}_\star\|_2^2,$$

*that is, the learned quantile under-covers by at least $\Omega(\|\widehat{\mathbf{w}} - \mathbf{w}_\star\|_2^2)$. Above, $c > 0$ is an absolute constant, and $r_0 > 0$ depends on $(\underline{\gamma}, \underline{\sigma}, K, \Phi_z, \alpha)$ but not $(n, d)$.*

**Implications; proof intuition** Theorem 3 shows that, for a broad class of data distributions, any estimator $\widehat{\mathbf{w}}$ will under-cover by at least $\Omega(\|\widehat{\mathbf{w}} - \mathbf{w}_\star\|_2^2)$. If particular, any estimator satisfying $\|\widehat{\mathbf{w}} - \mathbf{w}_\star\|_2 \asymp \widetilde{O}(\sqrt{d/n})$ (e.g. from standard generalization theory) will under-cover by $\widetilde{O}(d/n)$. This confirms that the estimation error in the (bulk) regression coefficient $\widehat{\mathbf{w}}$ is the primary source of the under-coverage bias, under assumptions that are more general than Theorem 1 in certain aspects (such as the distribution of $\mathbf{x}$ and $y|\mathbf{x}$). We remark that as opposed to Theorem 1, Therorem 3 does not give an end-to-end characterization of any specific algorithm, but assumes we have an estimator $\widehat{\mathbf{w}}$ with a small error.

At a high-level, Theorem 3 follows from the fact that any estimator $\widehat{\mathbf{w}}^\top \mathbf{x} + b_\star$ must be lower than the true quantile $\mathbf{w}_\star^\top \mathbf{x} + b_\star$ for some $\mathbf{x}$ and higher for some other $\mathbf{x}$. Averaging the coverage indicator over $\mathbf{x}$, such under-coverage and over-coverage cancel out on the first-order if $\mathbf{x}$ has a symmetric distribution, but aggregate to yield a *under-coverage effect on the second-order* as long as $\Phi_z(t)|_{t>0}$ *is concave* (which holds if $P_z$ is unimodal). The proof of Theorem 3 can be found in Appendix E.2.

## 5 Experiments

### 5.1 Simulations

**Setup** We first test our Theorem 1 via simulations. We generate data from the linear model (4) in $d = 100$ dimensions with $\|\mathbf{w}_\star\|_2 = 1$ and noise distribution $P_z = \mathsf{N}(0, 0.25)$. We vary $\kappa = d/n \in \{0.02, 0.04, \dots, 0.5\}$ where $\kappa$ determines a sample size $n$, and vary $\alpha \in \{0.5, 0.52, ..., 0.98\}$.

For each combination of $(\alpha, \kappa)$, we generate 8 random problem instances, and solve the quantile regression ERM problem (6) on each instance via (sub)-gradient descent. We evaluate the coverage of the learned quantile $\widehat{f}$ (thanks to the linear model (4), the coverage can be computed exactly without needing to introduce a test set). Additional details about the setup can be found in Appendix F.1.

**Results** Figure 1 plots the coverage of the learned quantiles. Observe that quantile regression exhibits under-coverage consistently across different values of $(\alpha, \kappa)$. Figure 1a shows that at fixed $\kappa$, the amount of under-coverage gets more severe at a higher $\alpha$, which is qualitatively consistent with our approximation formula $(\alpha - 1/2)\kappa$. Figure 1b further compares simulations with our analytical formula $\alpha - C_{\alpha, \kappa}$ (found numerically through solving the system of equations 12), as well as the

**Table 1:** Coverage (%) of quantile regression on real data at nominal level $\alpha = 0.9$. Each entry reports the test-set coverage with mean and std over 8 random seeds. $(d, n)$ denotes the {feature dim, # training examples}.

| Dataset | Linear | MLP-3-64 | MLP-3-512 | MLP-freeze-3-512 | $d$ | $n$ |
|---|---|---|---|---|---|---|
| Community | 88.63±1.53 | 76.46±1.41 | 63.09±2.91 | 87.85±1.30 | 100 | 1599 |
| Bike | 89.64±0.44 | 88.75±0.91 | 87.67±0.49 | 89.27±0.57 | 18 | 8708 |
| Star | 89.48±2.56 | 83.14±1.76 | 69.71±1.82 | 88.05±2.42 | 39 | 1728 |
| MEPS_19 | 90.09±0.72 | 85.46±0.96 | 78.55±0.93 | 89.03±0.51 | 139 | 12628 |
| MEPS_20 | 90.06±0.57 | 86.52±0.65 | 80.77±0.72 | 89.60±0.28 | 139 | 14032 |
| MEPS_21 | 89.99±0.39 | 83.79±0.52 | 73.09±0.82 | 89.15±0.36 | 139 | 12524 |
| Nominal ($\alpha$) | 90.00 | 90.00 | 90.00 | 90.00 | - | - |

local linear approximation $\alpha - (\alpha - 1/2)\kappa$ claimed in Theorem 1. Note that the simulations agree extremely well with the analytical formula. The approximation $\alpha - (\alpha - 1/2)\kappa$ is also very accurate for almost all $\kappa$ at $\alpha = 0.8$, and accurate for small $\kappa$ at $\alpha = 0.9, 0.95$. These verify our Theorem 1 and suggests it holds at rather realistic values of the dimension ($d = 100$).

## 5.2 Real data experiments

**Datasets and models** We take six real-world regression datasets: community and crimes (`Community`) [2], bike sharing (`Bike`) [1], Tennessee's student teacher achievement ratio (STAR) [6], as well as the medical expenditure survey number 19 (`MEPS_19`) [3], number 20 (`MEPS_20`) [4], and number 21 (`MEPS_21`) [5]. All datasets are pre-processed to have standarized features and randomly split into a 80% train set and 20% test set.

To go beyond linear quantile functions, we perform quantile regression with one of the following four models as our $f_\theta$: linear model (`Linear`), a 3-layer MLP (two non-linear layers) with width 64 (`MLP-3-64`), 512 (`MLP-3-512`), and a variant of the width-512 MLP where all representation layers are frozen and only the last linear layer is trained (`MLP-freeze-3-512`). All linear layers include a trainable bias. We minimize the $\alpha$-quantile loss (3) via momentum SGD with batch size 64. For each setting, we average over 8 random seeds where each seed determines the train-validation split, model initialization, and SGD batching. In our real experiments we fix $\alpha = 0.9$. (Results at $\alpha \in \{0.8, 0.95\}$ as well as additional experimental setups can be found in Appendix F.2).

**Results** Table 1 reports the coverage of the learned quantile functions (evaluated on the test sets). Observe that all MLPs exhibit under-coverage compared with the nominal level 90%. Additionally, the amount of under-coverage correlates well with model capacity—the two vanilla MLPs under-covers more severely than the MLP-freeze and the linear model. Notice that the linear model does not have a notable under-coverage on most datasets—we believe this is a consequence of $d/n$ being small on these datasets. The only exception is the `Community` dataset with the highest $d/n \approx 1/16$, on which the linear model does under-cover mildly by roughly 1%.

## 5.3 Linear quantile regression on pseudo-labels

To further test the coverage of linear quantile regression on real data distributions, we make two modifications: (1) We subset the training data by fixing $d$ and reducing $n$, so as to test the coverage across differerent values of $\kappa = d/n$; (2) We compare linear quantile regression on both true labels $y_i$, and *pseudo-labels* $y_i^{\text{pseudo}}$ generated from estimated linear models. These pseudo-labels are generated by first fitting a linear model $\widehat{\mathbf{w}} \in \mathbb{R}^d$ (with square loss) on the training data, and then generating a new label using the fitted linear model $\widehat{\mathbf{w}}$:

$$y_i^{\text{pseudo}} = \widehat{\mathbf{w}}^\top \mathbf{x}_i + \widehat{\sigma} z_i,$$

where $\widehat{\sigma}$ is estimated as $\sqrt{\widehat{\mathbb{E}}_{(\mathbf{x},y)}[(y - \widehat{\mathbf{w}}^\top \mathbf{x})^2]}$ on a separate hold-out split, and $z_i \sim \mathsf{N}(0, 1)$. The motivation for the pseudo-labels is to make sure that the data comes from a true linear model, removing the potential effect of model misspecification.

Table 2 shows that on the `MEPS_20` dataset, linear quantile regression exhibits under-coverage at relatively large values of $\kappa$ (0.1, 0.2, 0.5) for both kinds of labels. Also, there is no notable difference

between pseudo-labels and true labels. This provides evidence that our theory on *linear* quantile regression may hold broadly on real-world data distributions.

**Table 2:** Coverage of linear quantile regression on true labels vs. pseudo-labels.

| $\kappa = d/n$ | 0.01 | 0.02 | 0.05 | 0.1 | 0.2 | 0.5 |
|---|---|---|---|---|---|---|
| MEPS_20 | 89.83±0.67 | 89.89±0.81 | 89.54±0.82 | 88.74±1.51 | 87.15±1.52 | 84.75±1.81 |
| MEPS_20 Pseudo | 90.05±0.85 | 89.95±0.64 | 89.49±0.64 | 88.90±1.60 | 86.96±1.30 | 83.70±2.98 |
| Nominal ($\alpha$) | 90.00 | 90.00 | 90.00 | 90.00 | 90.00 | 90.00 |

# 6 Proof overview of Theorem 1

**Closed-form expression for coverage**   Our first step is to obtain a closed-form expression for the coverage. Recall that

$$\text{Coverage}(\widehat{f}) := \mathbb{P}_{(\mathbf{x},y)}(y \le \widehat{f}(\mathbf{x})) = \mathbb{P}_{(\mathbf{x},z)}(\langle \mathbf{w}_\star, \mathbf{x}\rangle + z \le \langle \widehat{\mathbf{w}}, \mathbf{x}\rangle + \widehat{b}).$$

As $\mathbf{x}$ is standard Gaussian, and the random variable $z$ has cumulative distribution function $\Phi_z$, standard calculation then yields the closed form expression (Lemma B.1)

$$\text{Coverage}(\widehat{f}) = \mathbb{E}_{G \sim \mathsf{N}(0,1)}[\Phi_z(\|\widehat{\mathbf{w}} - \mathbf{w}_\star\|_2 G + \widehat{b})].$$

**Concentration of $\|\widehat{\mathbf{w}} - \mathbf{w}_\star\|_2$ and $\widehat{b}$**   We generalize results from recent advances in high-dimensional M-estimator in linear models [23, 20, 31, 62] to show that $\|\widehat{\mathbf{w}} - \mathbf{w}_\star\|_2$ and $\widehat{b}$ obtained by quantile regression 6 concentrates around fixed values in the high-dimensional limit. We show that, in the limit of $d, n \to \infty$ and $d/n \to \kappa$, the following concentration happens:

$$\|\widehat{\mathbf{w}} - \mathbf{w}_\star\|_2 \xrightarrow{p} \tau_\star(\kappa), \quad \text{and} \quad \widehat{b} \xrightarrow{p} b_\star(\kappa). \tag{11}$$

Above, $\tau_\star$ and $b_\star$ are determined by the solutions of a system of nonlinear equations with three variables $(\tau, \lambda, b)$:

$$\begin{cases} \tau^2 \kappa = \lambda^2 \cdot \mathbb{E}_{(G,Z) \sim \mathsf{N}(0,1) \times P_z}[e'_{\ell_b^\alpha}(\tau G + Z; \lambda)^2], \\ \tau \kappa = \lambda \cdot \mathbb{E}_{(G,Z) \sim \mathsf{N}(0,1) \times P_z}[e'_{\ell_b^\alpha}(\tau G + Z; \lambda)G], \\ 0 = \mathbb{E}_{(G,Z) \sim \mathsf{N}(0,1) \times P_z}[e'_{\ell_b^\alpha}(\tau G + Z; \lambda)], \end{cases} \tag{12}$$

where $e_\ell(x; \tau) = \min_v \frac{1}{2\tau}(x - v)^2 + \ell(v)$ and $\ell_b^\alpha = \ell^\alpha(t - b)$ is the shifted pinball loss (2). (See Theorem C.1 for the formal statement.) This is established via two main steps: We first build on the results of Thrampoulidis et al. [62] to show that a variant of the risk minimization problem with a fixed bias $b$ concentrates around the solution to the first two equations in (12). We then develop a novel concentration argument to deal with the additional learnable bias $b$ in the minimization problem (6), which introduces the third equation in (12) that will be used in characterizing the limiting value of the minimizer $\widehat{b}$.

The concentration (11) implies that $\text{Coverage}(\widehat{f})$ also converges to the following limiting coverage value (Lemma C.3):

$$\text{Coverage}(\widehat{f}) \xrightarrow{p} \mathbb{E}_{G \sim \mathsf{N}(0,1)}[\Phi_z(\tau_\star(\kappa)G + b_\star(\kappa))] =: \alpha - C_{\alpha,\kappa}. \tag{13}$$

**Calculating the limiting coverage via local linear analysis**   In this final step, as another technical crux of the proof, we further evaluate the small $\kappa$ approximation of coverage value (13), and determine the sign of $C_{\alpha,\kappa}$. This is achieved by a *local linear analysis* on the solutions of the aforementioned system of equations at small $\kappa$ (Lemma C.2) in a similar fashion as in [8], and a precise analysis on the interplay between the concentration values $\tau_\star$, $b_\star$, and the noise density $\phi_z$. Combining these calculations yields that $C_{\alpha,\kappa}/\kappa = -(\alpha - 1/2) + o(1)$ for small enough $\kappa$ (Lemma C.4). As $\alpha > 1/2$, this establishes Theorem 1. All details on these analyses can be found in our proofs in Appendix C.

# 7 Conclusion

This paper presents a first theoretical justification of the under-coverage bias in quantile regression. We prove that quantile regression suffers from an inherent under-coverage bias even in well-specified linear settings, and provide a precise quantitative characterization of the amount of the under-coverage bias on Gaussian linear models. Our theory further identifies the high-dimensional estimation error in the regression coefficient as the main source of this under-coverage bias, which holds more generally on a broad class of data distributions. We believe our work opens up several interesting directions for future work, such as analyzing non-linear quantile regression, as well as analyzing other notions of uncertainty in regression problems.

## Funding transparency statement

YB, HW, CX are funded through employment with Salesforce.

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
