# A   Technical tools

## A.1   The pinball loss

Recall that we took $\ell^\alpha : \mathbb{R} \to \mathbb{R}_{\geq 0}$ to be the pinball loss for the $\alpha$-quantile, i.e.,

$$\ell^\alpha(t) = -(1-\alpha)t\mathbf{1}\{t \leq 0\} + \alpha t\mathbf{1}\{t > 0\}.$$

We denote $\ell_b^\alpha(t) = \ell^\alpha(t-b)$ to be the shifted pinball loss. We will suppress the superscript in $\ell_b = \ell_b^\alpha$ whenever it is clear in the context. The loss function $\ell_b$ is weakly differentiable, with a weak derivative $\ell_b'$ given by

$$\ell_b'(t) = -(1-\alpha)\mathbf{1}\{t \leq 0\} + \alpha\mathbf{1}\{t > 0\}.$$

## A.2   Calculus of the Moreau envelope and prox operator

Given a convex loss function $\ell : \mathbb{R} \to \mathbb{R}$, we define its the Moreau envelope $e_\ell : \mathbb{R} \times \mathbb{R}_{>0} \to \mathbb{R}$ by

$$e_\ell(x; \lambda) = \min_v \left[\frac{1}{2\lambda}(x-v)^2 + \ell(v)\right],$$

and the proximal operator $\mathsf{prox}_\ell(x; \lambda) : \mathbb{R} \times \mathbb{R}_{>0} \to \mathbb{R}$ by

$$\mathsf{prox}_\ell(x; \lambda) = \arg\min_v \left[\frac{1}{2\lambda}(x-v)^2 + \ell(v)\right].$$

Since $\ell$ is convex, $\mathsf{prox}_\ell(x; \lambda)$ is well-defined. For $\ell = \ell_b$, we have

$$\mathsf{prox}_{\ell_b}(x; \lambda) = b \cdot \mathbf{1}\{x \in [b - (1-\alpha)\lambda, b + \alpha\lambda]\}$$
$$+ (x - \alpha\lambda)\mathbf{1}\{x > b + \alpha\lambda\} + (x + (1-\alpha)\lambda)\mathbf{1}\{x < b - (1-\alpha)\lambda\}.$$

The function $e_{\ell_b}$ is differentiable with respect to $(x, \lambda, b)$, with derivatives

$$\partial_x e_{\ell_b}(x; \lambda) = \frac{x - \mathsf{prox}_{\ell_b}(x; \lambda)}{\lambda},$$
$$\partial_\lambda e_{\ell_b}(x; \lambda) = -\frac{[x - \mathsf{prox}_{\ell_b}(x; \lambda)]^2}{2\lambda^2} = -\frac{1}{2}(\partial_x e_{\ell_b(x;\lambda)})^2, \tag{14}$$
$$\partial_b e_{\ell_b}(x; \lambda) = -\partial_x e_{\ell_b}(x; \lambda).$$

The functions $\partial_x e_{\ell_b}$, $\partial_\lambda e_{\ell_b}$ and $\partial_b e_{\ell_b}$ are weakly-differentiable with respect to $(x, \lambda, b)$, with the following formulas giving one (choice of) weak derivative:

$$\partial_x \partial_x e_{\ell_b}(x; \lambda) = \frac{1}{\lambda}\mathbf{1}\{\mathsf{prox}_{\ell_b}(x; \lambda) = b\} \geq 0,$$
$$\partial_\lambda \partial_x e_{\ell_b}(x; \lambda) = -\frac{[x - \mathsf{prox}_{\ell_b}(x; \lambda)]^2}{2\lambda^2} = -\partial_x e_{\ell_b}(x; \lambda)\partial_x \partial_x e_{\ell_b}(x; \lambda),$$
$$\partial_b \partial_x e_{\ell_b}(x; \lambda) = -\partial_x \partial_x e_{\ell_b}(x; \lambda), \tag{15}$$
$$\partial_\lambda \partial_b e_{\ell_b}(x; \lambda) = -\partial_x e_{\ell_b}(x; \lambda)\partial_b \partial_x e_{\ell_b}(x; \lambda) = \partial_x e_{\ell_b}(x; \lambda)\partial_x \partial_x e_{\ell_b}(x; \lambda),$$
$$\partial_b \partial_b e_{\ell_b}(x; \lambda) = \partial_x \partial_x e_{\ell_b}(x; \lambda),$$
$$\partial_\lambda \partial_\lambda e_{\ell_b}(x; \lambda) = -\partial_x e_{\ell_b}(x; \lambda)\partial_\lambda \partial_x e_{\ell_b}(x; \lambda) = \partial_x e_{\ell_b}(x; \lambda)^2 \partial_x \partial_x e_{\ell_b}(x; \lambda).$$

## A.3   Implicit function theorem

We state the standard implicit function theorem in the following.

**Lemma A.1** (Implicit function theorem). *Let $\boldsymbol{F}(\boldsymbol{p}, \kappa) : \mathbb{R}^s \times \mathbb{R}_{\geq 0} \to \mathbb{R}^s$ be a continuously differentiable vector-valued function on $\mathsf{B}(\boldsymbol{p}_0, \varepsilon) \times [0, \bar{\kappa}_0)$ for some $\bar{\kappa}_0 > 0$. Suppose $\boldsymbol{F}(\boldsymbol{p}_0, 0) = 0$ and*

$$\sigma_{\min}(\nabla_{\boldsymbol{p}} F(\boldsymbol{p}_0, 0)) > 0.$$

*Then there exists a constant $\kappa_0 > 0$ and a continuous differentiable path $\boldsymbol{p}_\star(\kappa) \in \mathsf{B}(\boldsymbol{p}_0, \varepsilon)$, such that*

$$\boldsymbol{F}(\boldsymbol{p}_\star(\kappa), \kappa) = 0, \quad \forall \kappa \in [0, \kappa_0).$$

## A.4 Other technical lemmas

**Lemma A.2.** *For any vectors $\mathbf{u}, \mathbf{v} \in \mathbb{R}^d$ and any positive definite matrix $\mathbf{A} \in \mathbb{R}^{d \times d}$, $\mathbf{A} \succ 0$, we have*

$$\left| \mathbf{u}^\top (\mathbf{A} + \mathbf{v}\mathbf{v}^\top)^{-1} \mathbf{v} \right| \leq \left| \mathbf{u}^\top \mathbf{A}^{-1} \mathbf{v} \right|.$$

*Proof.* Recall the Sherman-Morrison-Woodbury identity for matrix inversion:

$$(\mathbf{A} + \mathbf{v}\mathbf{v}^\top)^{-1} = \mathbf{A}^{-1} - \frac{\mathbf{A}^{-1}\mathbf{v}\mathbf{v}^\top \mathbf{A}^{-1}}{1 + \mathbf{v}^\top \mathbf{A}^{-1}\mathbf{v}}.$$

Applying this, we have

$$\left| \mathbf{u}^\top (\mathbf{A} + \mathbf{v}\mathbf{v}^\top)^{-1} \mathbf{v} \right| = \left| \mathbf{u}^\top \mathbf{A}^{-1}\mathbf{v} - \mathbf{u}^\top \frac{\mathbf{A}^{-1}\mathbf{v}\mathbf{v}^\top \mathbf{A}^{-1}}{1 + \mathbf{v}^\top \mathbf{A}^{-1}\mathbf{v}} \mathbf{v} \right|$$

$$= \left| \mathbf{u}^\top \mathbf{A}^{-1}\mathbf{v} - (\mathbf{u}^\top \mathbf{A}^{-1}\mathbf{v}) \cdot \frac{\mathbf{v}^\top \mathbf{A}^{-1}\mathbf{v}}{1 + \mathbf{v}^\top \mathbf{A}^{-1}\mathbf{v}} \right|$$

$$= \left| (\mathbf{u}^\top \mathbf{A}^{-1}\mathbf{v}) \cdot \frac{1}{1 + \mathbf{v}^\top \mathbf{A}^{-1}\mathbf{v}} \right| \leq \left| \mathbf{u}^\top \mathbf{A}^{-1}\mathbf{v} \right|.$$

Above, the last line used $\mathbf{v}^\top \mathbf{A}^{-1}\mathbf{v} \geq 0$ since $\mathbf{A}^{-1} \succeq 0$. This proves the lemma. $\qquad \square$

**Lemma A.3.** *Let $X \in \mathbb{R}^s$ be a random variable with distribution $\mu$, and let $u : \mathbb{R}^s \to \mathbb{R}^k$ be a continuous function. Assume that there exist $(x_t)_{t \in [k]}$ that are in the support of the distribution of $X$ (i.e., for any $t \in [k]$, we have $\mu(\{x : \|x_t - x\|_2 \leq \varepsilon\}) > 0$ for any $\varepsilon > 0$), such that $[u(x_1), \ldots, u(x_k)] \in \mathbb{R}^{k \times k}$ is full rank. Then we have*

$$\mathbb{E}[u(X)u(X)^\top] \succ 0.$$

*Proof of Lemma A.3.* We denote

$$\omega(\varepsilon) = \sup_{t \in [k]} \left[ 2 \sup_{x \in \mathsf{B}(x_t, \varepsilon)} \|u(x) - u(x_t)\|_2 \cdot \sup_{x \in \mathsf{B}(x_t, \varepsilon)} \|u(x)\|_2 + \sup_{x \in \mathsf{B}(x_t, \varepsilon)} \|u(x) - u(x_t)\|_2^2 \right].$$

Since $u$ is a continuous function on $\mathbb{R}^s$, we have

$$\lim_{\varepsilon \to 0} \omega(\varepsilon) = 0.$$

We further denote

$$\nu(\varepsilon) = \min_{t \in [k]} \mu(\mathsf{B}(x_t, \varepsilon)).$$

Then by the fact that $(x_t)_{t \in [k]} \subseteq \operatorname{supp}(\mu)$, we have $\nu(\varepsilon) > 0$ for any $\varepsilon > 0$.

Then, for any $\varepsilon > 0$, we have

$$\mathbb{E}[u(X)u(X)^\top] \succeq \sum_{t=1}^{k} \int_{\mathsf{B}(x_t, \varepsilon)} u(x)u(x)^\top \mu(\mathrm{d}x)$$

$$\succeq \sum_{t=1}^{k} (u(x_t)u(x_t)^\top - \omega(\varepsilon)I_k)\nu(\varepsilon)$$

$$= \nu(\varepsilon) \sum_{t=1}^{k} u(x_t)u(x_t)^\top - \omega(\varepsilon)k\nu(\varepsilon)I_k$$

$$\succeq \nu(\varepsilon) \left[ \lambda_{\min}\left( \sum_{t=1}^{k} u(x_t)u(x_t)^\top \right) - \omega(\varepsilon)k \right] I_k.$$

Since $[u(x_1), \ldots, u(x_k)]$ has full rank, we have $\lambda_{\min}(\sum_{t=1}^{k} u(x_t)u(x_t)^\top) > 0$. We can choose $\varepsilon$ sufficiently small, so that $\lambda_{\min}(\sum_{t=1}^{k} u(x_t)u(x_t)^\top) - \omega(\varepsilon)k > 0$. This gives $\mathbb{E}[u(X)u(X)^\top] \succ 0$. This proves the lemma. $\qquad \square$

# B    Properties of quantile regression

## B.1    Population minimizer of quantile risk

We can express the population quantile risk as
$$R(f) = \mathbb{E}[\ell^\alpha(y - f(\mathbf{x}))] = \mathbb{E}_{\mathbf{x}}\mathbb{E}[\ell^\alpha(y - f(\mathbf{x}))|\mathbf{x}].$$
Therefore, any function $f(\mathbf{x})$ that minimizes the conditional expectation $\mathbb{E}[\ell^\alpha(y - f(\mathbf{x}))|\mathbf{x}]$ at every $\mathbf{x}$ minimizes the above risk. It is a classical result that for any distribution $P$ on $\mathbb{R}$, a minimizer of $\mathbb{E}_{y \sim P}[\ell^\alpha(y - f)]$ is the $\alpha$-quantile $q_\alpha = \inf\{t \in \mathbb{R} : F(t) \geq \alpha\}$, where $F$ is the CDF of $P$ [34, Section 3]. Therefore, the conditional quantile function $q^\star(\mathbf{x}) = \arg\min_f \mathbb{E}[\ell^\alpha(y - f(\mathbf{x}))|\mathbf{x}]$ is a minimizer of the aforementioned conditional expectation at every $\mathbf{x}$. This proves the claim.    □

## B.2    Explicit expression of coverage

**Lemma B.1.** *Under the linear model* (4)*, for any linear quantile function* $\widehat{f}(\mathbf{x}) = \widehat{\mathbf{w}}^\top\mathbf{x} + \widehat{b}$*, the coverage of* $\widehat{f}$ *can be expressed as*
$$\mathrm{Coverage}(\widehat{f}) = \mathbb{P}_{(\mathbf{x},y)}\left(y \leq \widehat{\mathbf{w}}^\top\mathbf{x} + \widehat{b}\right) = \mathbb{E}_{G \sim \mathsf{N}(0,1)}\left[\Phi_z\left(\|\widehat{\mathbf{w}} - \mathbf{w}_\star\|_2 \, G + \widehat{b}\right)\right].$$

*Proof.* By the linear model (4), we have $y = \mathbf{w}_\star^\top\mathbf{x} + z$ and thus
$$\begin{aligned}
\mathbb{P}_{(\mathbf{x},y)}\left(y \leq \widehat{\mathbf{w}}^\top\mathbf{x} + \widehat{b}\right) &= \mathbb{P}_{(\mathbf{x},z)}\left(\mathbf{w}_\star^\top\mathbf{x} + z \leq \widehat{\mathbf{w}}^\top\mathbf{x} + \widehat{b}\right) \\
&= \mathbb{P}_{(\mathbf{x},z)}\left(z \leq (\widehat{\mathbf{w}} - \mathbf{w}_\star)^\top\mathbf{x} + \widehat{b}\right) \\
&= \mathbb{E}_{\mathbf{x}}\left[\Phi_z\left((\widehat{\mathbf{w}} - \mathbf{w}_\star)^\top\mathbf{x} + \widehat{b}\right)\right] \\
&= \mathbb{E}_{G \sim \mathsf{N}(0,1)}\left[\Phi_z\left(\|\widehat{\mathbf{w}} - \mathbf{w}_\star\|_2 \, G + \widehat{b}\right)\right].
\end{aligned}$$
Above, the last step used the Gaussian input assumption $\mathbf{x} \sim \mathsf{N}(\mathbf{0}, \mathbf{I}_d)$.    □

# C    Proof of Theorem 1

Recall that $\ell_b^\alpha(t) = \ell^\alpha(t - b)$ where $\ell^\alpha(t)$ is the pinball loss for the $\alpha$-quantile, i.e.,
$$\ell^\alpha(t) = -(1 - \alpha)t\mathbf{1}\{t \leq 0\} + \alpha t\mathbf{1}\{t > 0\}.$$
We will consider a fixed $\alpha$, so we often write $\ell_b \equiv \ell_b^\alpha$. We further define
$$e_\ell(x; \lambda) := \min_{v \in \mathbb{R}}\left[\frac{1}{2\lambda}(x - v)^2 + \ell(v)\right].$$
We consider the following system of equations in three variables $(\tau, \lambda, b) \in \mathbb{R}_{>0} \times \mathbb{R}_{>0} \times \mathbb{R}$, which will be key to our analysis of the quantile ERM problem (6):
$$\begin{cases}
\tau^2\kappa = \lambda^2 \cdot \mathbb{E}\left[e'_{\ell_b}(\tau G + Z; \lambda)^2\right], \\
\tau\kappa = \lambda \cdot \mathbb{E}\left[e'_{\ell_b}(\tau G + Z; \lambda)G\right], \\
0 = \mathbb{E}\left[e'_{\ell_b}(\tau G + Z; \lambda)\right].
\end{cases} \tag{16}$$

The following two lemmas show that the system of equations (16) has a unique solution, which further admits a local linear expansion over $\kappa$ with closed-form coefficients.

**Lemma C.1** (Existence of unique solution). *There exists $\kappa_0 > 0$ such that for any $\kappa \in (0, \kappa_0]$, there exists a unique solution $(\tau_\star(\kappa), \lambda_\star(\kappa), b_\star(\kappa))$ of the system of equations* (16)*.*

Define constants
$$\begin{aligned}
\bar{\tau}_0^2 &:= \frac{\alpha(1 - \alpha)}{\phi_z^2(z_\alpha)}, \\
\bar{\lambda}_0 &:= \frac{1}{\phi_z(z_\alpha)}, \\
\bar{b}_0 &:= \frac{-\alpha(1 - \alpha)\phi_z'(z_\alpha) - (2\alpha - 1)\phi_z^2(z_\alpha)}{2\phi_z^3(z_\alpha)}.
\end{aligned} \tag{17}$$

**Lemma C.2** (Local linear expansion of solution at small $\kappa$). *Let $(\tau_\star(\kappa), \lambda_\star(\kappa), b_\star(\kappa))$ denote the solutions to (16) for any $\kappa \in (0, \kappa_0]$. The following local linear expansion holds at small $\kappa$:*

$$
\begin{aligned}
\tau_\star^2(\kappa) &= \bar{\tau}_0^2 \kappa + o(\kappa), \\
\lambda_\star(\kappa) &= \bar{\lambda}_0 \kappa + o(\kappa), \\
b_\star(\kappa) &= z_\alpha + \bar{b}_0 \kappa + o(\kappa),
\end{aligned}
\tag{18}
$$

*where $z_\alpha = \Phi_z^{-1}(\alpha)$ is the $\alpha$-quantile of $P_z$.*

We now show that the quantile ERM problem (6) exhibits a sharp concentration in the proportional limit ($n, d \to \infty$, $d/n \to \kappa$) where the concentration values are determined by the solutions $(\tau_\star^2(\kappa), \lambda_\star(\kappa), b_\star(\kappa))$ above. This result is a novel extension of (the unregularized case of) [62, Theorem 4.1] in that it incorporates—and proves the concentration in presence of—the additional trainable bias parameter $b$. Recall the ERM problem (6) is

$$
(\widehat{\mathbf{w}}, \widehat{b}) \in \arg\min_{\mathbf{w}, b} \widehat{R}_n(\mathbf{w}, b) := \frac{1}{n} \sum_{i=1}^n \ell^\alpha(y_i - (\mathbf{w}^\top \mathbf{x}_i + b)).
\tag{19}
$$

**Theorem C.1** (Concentration of quantile ERM). *Under the linear model (4) and Assumption A, consider the limit $n, d \to \infty$ and $d/n \to \kappa \in (0, \kappa_0]$ where $\kappa_0 > 0$ is some constant. Then with probability approaching one, the empirical risk minimizer $(\widehat{\mathbf{w}}, \widehat{b})$ exists (but may not be unique), and for any empirical risk minimizer $(\widehat{\mathbf{w}}, \widehat{b})$, we have*

$$
\widehat{b} \overset{p}{\to} b_\star(\kappa), \quad \|\widehat{\mathbf{w}} - \mathbf{w}_\star\|_2^2 \overset{p}{\to} \tau_\star^2(\kappa).
$$

Denote

$$
\text{Coverage}_{\alpha, \kappa} \equiv \mathbb{E}_{G \sim \mathsf{N}(0,1)}[\Phi_z(\tau_\star(\kappa)G + b_\star(\kappa))].
$$

Combining Theorem C.1, Lemma C.2, and the expression of the coverage in Lemma B.1, the following two lemmas show that $\text{Coverage}(\widehat{f})$ also concentrates around a value $\text{Coverage}_{\alpha, \kappa} = \alpha - C_{\alpha, \kappa}$, where $C_{\alpha, \kappa}$ admits a local linear expansion with a closed-form coefficient.

**Lemma C.3.** *Under the settings of Theorem 1, we have as $n, d \to \infty$, $d/n \to \kappa \in (0, \kappa_0]$,*

$$
\text{Coverage}(\widehat{f}) \overset{p}{\to} \text{Coverage}_{\alpha, \kappa}.
\tag{20}
$$

**Lemma C.4.** *Under the same setting as Lemma C.3, we further have*

$$
\begin{aligned}
\text{Coverage}_{\alpha, \kappa} &= \alpha - C_{\alpha, \kappa} \\
&= \alpha + (\phi_z(z_\alpha)\bar{b}_0 + (1/2)\phi_z'(z_\alpha)\bar{\tau}_0^2)\kappa + o(\kappa).
\end{aligned}
\tag{21}
$$

By Lemma C.4 and the definition of $\bar{b}_0$ and $\bar{\tau}_0^2$ in (17), the above coefficient in front of $\kappa$ can be simplified as

$$
\begin{aligned}
&\phi_z(z_\alpha)\bar{b}_0 + (1/2)\phi_z'(z_\alpha)\bar{\tau}_0^2 \\
&= \phi_z(z_\alpha) \cdot \frac{-\alpha(1-\alpha)\phi_z'(z_\alpha) - (2\alpha - 1)\phi_z^2(z_\alpha)}{2\phi_z^3(z_\alpha)} + \frac{1}{2}\phi_z'(z_\alpha) \cdot \frac{\alpha(1-\alpha)}{\phi_z^2(z_\alpha)} \\
&= -(\alpha - 1/2).
\end{aligned}
$$

This shows that $C_{\alpha, \kappa} = (\alpha - 1/2)\kappa + o(\kappa)$, and in particular $C_{\alpha, \kappa} > 0$ for all small $\kappa$ as $\alpha - 1/2 > 0$. This proves Theorem 1. $\qquad \square$

The rest of this section is organized as follows. We prove Lemma C.1 in Section C.1 (which requires analyzing a transformed system of equations and applying the implicit function theorem). In Section C.2, we connect the system of equations to a variational problem over four real variables. We then use this connection to prove Theorem C.1 in Section C.3. Finally, we prove Lemma C.3 and Lemma C.4 in Section C.4.

## C.1 Proof of Lemma C.1 and Lemma C.2

### C.1.1 Analysis of system of equations (16)

We first perform a change of variables. For any $(\bar{\tau}, \bar{\lambda}, \bar{b}, \kappa) \in \overline{\Omega} \times (0, 1)$ where $\overline{\Omega} = \mathbb{R}_{\geq 0} \times \mathbb{R}_{\geq 0} \times \mathbb{R}$, we rewrite the system of equations (16) as

$$\boldsymbol{F}(\boldsymbol{p}; \kappa) = \boldsymbol{0}, \tag{22}$$

where $\boldsymbol{p} = (\bar{\tau}, \bar{\lambda}, \bar{b})$, $\boldsymbol{F}(\boldsymbol{p}; \kappa) := (F_1(\boldsymbol{p}; \kappa), F_2(\boldsymbol{p}; \kappa), F_3(\boldsymbol{p}; \kappa))$ in which

$$
\begin{aligned}
F_1(\bar{\tau}, \bar{\lambda}, \bar{b}; \kappa) &:= \bar{\tau}^2 - \bar{\lambda}^2 \cdot \mathbb{E}\left[e'_{\ell_{\bar{b}\kappa + z_\alpha}}(\bar{\tau}\sqrt{\kappa}G + Z; \bar{\lambda}\kappa)^2\right], \\
F_2(\bar{\tau}, \bar{\lambda}, \bar{b}; \kappa) &:= \bar{\tau} - \kappa^{-1/2}\bar{\lambda} \cdot \mathbb{E}\left[e'_{\ell_{\bar{b}\kappa + z_\alpha}}(\bar{\tau}\sqrt{\kappa}G + Z; \bar{\lambda}\kappa)G\right], \\
F_3(\bar{\tau}, \bar{\lambda}, \bar{b}; \kappa) &:= \kappa^{-1}\mathbb{E}\left[e'_{\ell_{\bar{b}\kappa + z_\alpha}}(\bar{\tau}\sqrt{\kappa}G + Z; \bar{\lambda}\kappa)\right].
\end{aligned}
\tag{23}
$$

Equation (22) and the system (16) are equivalent up to a change of variables: For any fixed $\kappa$, any solution $(\tau_\star, \lambda_\star, b_\star)$ of Eq. (16) yields a solution $(\tau_\star/\kappa, \lambda_\star/\kappa, (b_\star - z_\alpha)/\kappa, \kappa)$ of $\boldsymbol{F}(\boldsymbol{p}; \kappa) = \boldsymbol{0}$, and vice versa. Notice that this equivalence allows us to establish Lemma C.1 and Lemma C.2 by considering the transformed equation (22).

The following two auxiliary lemmas, which give a continuity analysis of the function $\boldsymbol{F}$, are key to establishing Lemma C.1 and Lemma C.2. These auxiliary lemmas are required for checking the conditions of the implicit function theorem. The proofs of these two lemmas are deferred to Section C.1.2 and C.1.3 respectively. As a shorthand, we take

$$\boldsymbol{p}_0 = (\bar{\tau}_0, \bar{\lambda}_0, \bar{b}_0),$$

where $\bar{\tau}_0, \bar{\lambda}_0, \bar{b}_0$ are defined in (17).

**Lemma C.5.** *Let Assumption A hold. Let $\boldsymbol{F}$ be as defined in Eq. (22). Then for any $\varepsilon$ such that $\mathsf{B}(\boldsymbol{p}_0, 2\varepsilon) \subseteq \overline{\Omega} = \mathbb{R}_{\geq 0} \times \mathbb{R}_{\geq 0} \times \mathbb{R}$, there exists a continuous matrix function $\boldsymbol{J} : \mathsf{B}(\boldsymbol{p}_0, \varepsilon) \to \mathbb{R}^{3 \times 3}$ with*

$$\sigma_{\min}(\boldsymbol{J}(\boldsymbol{p}_0)) > 0, \tag{24}$$

*and*

$$\lim_{\kappa \to 0} \sup_{\boldsymbol{p} \in \mathsf{B}(\boldsymbol{p}_0, \varepsilon)} \left\|\nabla_{\boldsymbol{p}} \boldsymbol{F}(\boldsymbol{p}, \kappa) - \boldsymbol{J}(\boldsymbol{p})\right\|_{\mathrm{op}} = 0. \tag{25}$$

**Lemma C.6.** *Let Assumption A hold. Let $\boldsymbol{F}$ be as defined in Eq. (22). Then for any $\varepsilon$ such that $\mathsf{B}(\boldsymbol{p}_0, 2\varepsilon) \subseteq \overline{\Omega} = \mathbb{R}_{\geq 0} \times \mathbb{R}_{\geq 0} \times \mathbb{R}$, there exists two continuous vector functions $\boldsymbol{F}_0, \boldsymbol{g} : \mathsf{B}(\boldsymbol{p}_0, \varepsilon) \to \mathbb{R}^3$ such that*

$$\lim_{\kappa \to 0} \sup_{\boldsymbol{p} \in \mathsf{B}(\boldsymbol{p}_0, \varepsilon)} \left\|\boldsymbol{F}(\boldsymbol{p}, \kappa) - \boldsymbol{F}_0(\boldsymbol{p})\right\|_2 = 0,$$

$$\lim_{\kappa \to 0} \sup_{\boldsymbol{p} \in \mathsf{B}(\boldsymbol{p}_0, \varepsilon)} \left\|\partial_\kappa \boldsymbol{F}(\boldsymbol{p}, \kappa) - \boldsymbol{g}(\boldsymbol{p})\right\|_2 = 0.$$

*Moreover, we have*

$$\lim_{\kappa \to 0+} \boldsymbol{F}(\boldsymbol{p}_0, \kappa) = \boldsymbol{F}_0(\boldsymbol{p}_0) = \boldsymbol{0}.$$

By Lemma C.5 and C.6, we can continuously extend the function $\boldsymbol{F}$ to the region $\mathsf{B}(\boldsymbol{p}_0, \varepsilon) \times [0, \kappa_0)$ for some small $\kappa_0$, such that $\boldsymbol{F}(\boldsymbol{p}, \kappa)$ is continuously differentiable in the same region. Moreover, by Lemma C.6, we have $\boldsymbol{F}(\boldsymbol{p}_0, 0) = \lim_{\kappa \to 0} \boldsymbol{F}(\boldsymbol{p}_0, \kappa) = 0$. Finally, by Lemma C.5, we have $\sigma_{\min}(\nabla_{\boldsymbol{p}} \boldsymbol{F}(\boldsymbol{p}_0, 0)) > 0$.

### C.1.2 Proof of Lemma C.5

For any $\boldsymbol{p} = (\bar{\tau}, \bar{\lambda}, \bar{b}) \in \overline{\Omega} = \mathbb{R}_{\geq 0} \times \mathbb{R}_{\geq 0} \times \mathbb{R}$, we define a continuous matrix function $\boldsymbol{J} : \overline{\Omega} \to \mathbb{R}^{3 \times 3}$ by

$$
\boldsymbol{J}(\boldsymbol{p}) = \begin{pmatrix}
2\bar{\tau} & -2\bar{\lambda}\alpha(1-\alpha) & 0 \\
1 - \bar{\lambda}\phi_z(z_\alpha) & -\bar{\tau}\phi_z(z_\alpha) & 0 \\
-\bar{\tau}\phi'_z(z_\alpha) & (1 - 2\alpha)\phi_z(z_\alpha) & -\phi_z(z_\alpha)
\end{pmatrix}.
$$

Evaluating $\boldsymbol{J}(\boldsymbol{p}_0)$ (recall $\boldsymbol{p}_0$ is defined in Eq. (17)), we have

$$
\boldsymbol{J}(\boldsymbol{p}_0) = \begin{pmatrix} \frac{2\sqrt{\alpha(1-\alpha)}}{\phi_z(z_\alpha)} & -\frac{2\alpha(1-\alpha)}{\phi_z(z_\alpha)} & 0 \\ 0 & -\sqrt{\alpha(1-\alpha)} & 0 \\ -\frac{\sqrt{\alpha(1-\alpha)}}{\phi_z(z_\alpha)}\phi_z'(z_\alpha) & (1-2\alpha)\phi_z(z_\alpha) & -\phi_z(z_\alpha) \end{pmatrix}.
$$

Since we have assumed that $\phi_z(z_\alpha) \neq 0$, it is easy to see that $\det(\boldsymbol{J}(\boldsymbol{p}_0)) = -2\alpha(1-\alpha) \neq 0$. This proves Eq. (24).

We next prove Eq. (25). Recall that the definition of $\boldsymbol{F} = (F_1, F_2, F_3)$ as given in Eq. (23), by the calculus of $e_{\ell_b}$ as in Section A.2, we have

$$
F_1(\boldsymbol{p}; \kappa) = \bar{\tau}^2 - \mathbb{E}_G\Big\{ \frac{1}{\kappa^2} \int_{[\overline{G}_-, \overline{G}_+]} (z - \overline{G})^2 \phi_z(z)\mathrm{d}z + \bar{\lambda}^2\alpha^2[1 - \Phi_z(\overline{G}_+)] + \bar{\lambda}^2(1-\alpha)^2\Phi_z(\overline{G}_-) \Big\},
$$

$$
F_2(\boldsymbol{p}; \kappa) = \bar{\tau} - \kappa^{-1/2}\mathbb{E}_G\Big\{ \frac{1}{\kappa} \int_{[\overline{G}_-, \overline{G}_+]} (z - \overline{G})G\phi_z(z)\mathrm{d}z + \bar{\lambda}\alpha[(1 - \Phi_z(\overline{G}_+))G] - \bar{\lambda}(1-\alpha)\Phi_z(\overline{G}_-)G \Big\},
$$

$$
F_3(\boldsymbol{p}; \kappa) = \kappa^{-1}\mathbb{E}_G\Big\{ \frac{1}{\bar{\lambda}\kappa} \int_{[\overline{G}_-, \overline{G}_+]} (z - \overline{G})\phi_z(z)\mathrm{d}z + \alpha[1 - \Phi_z(\overline{G}_+)] - (1-\alpha)\Phi_z(\overline{G}_-) \Big\},
$$

where

$$
\begin{aligned}
\overline{G} &\equiv z_\alpha + \kappa\bar{b} - G\bar{\tau}\sqrt{\kappa}, \\
\overline{G}_+ &\equiv z_\alpha + \kappa\bar{b} + \alpha\kappa\bar{\lambda} - G\bar{\tau}\sqrt{\kappa}, \\
\overline{G}_- &\equiv z_\alpha + \kappa\bar{b} - (1-\alpha)\kappa\bar{\lambda} - G\bar{\tau}\sqrt{\kappa}.
\end{aligned}
\tag{26}
$$

Using the smoothness property of $\phi_z$, with some calculus, we have

$$
\begin{aligned}
\lim_{\kappa\to 0} \partial_{\bar{\tau}}F_1(\boldsymbol{p}; \kappa) &= 2\bar{\tau}, \\
\lim_{\kappa\to 0} \partial_{\bar{\tau}}F_2(\boldsymbol{p}; \kappa) &= 1 - \bar{\lambda}\phi_z(z_\alpha), \\
\lim_{\kappa\to 0} \partial_{\bar{\tau}}F_3(\boldsymbol{p}; \kappa) &= -\bar{\tau}\phi_z'(z_\alpha), \\
\lim_{\kappa\to 0} \partial_{\bar{\lambda}}F_1(\boldsymbol{p}; \kappa) &= -2\bar{\lambda}\alpha(1-\alpha), \\
\lim_{\kappa\to 0} \partial_{\bar{\lambda}}F_2(\boldsymbol{p}; \kappa) &= -\bar{\tau}\phi_z(z_\alpha), \\
\lim_{\kappa\to 0} \partial_{\bar{\lambda}}F_3(\boldsymbol{p}; \kappa) &= (1-2\alpha)\phi_z(z_\alpha), \\
\lim_{\kappa\to 0} \partial_{\bar{b}}F_1(\boldsymbol{p}; \kappa) &= 0, \\
\lim_{\kappa\to 0} \partial_{\bar{b}}F_2(\boldsymbol{p}; \kappa) &= 0, \\
\lim_{\kappa\to 0} \partial_{\bar{b}}F_3(\boldsymbol{p}; \kappa) &= -\phi_z(z_\alpha).
\end{aligned}
$$

This proves that $\lim_{\kappa\to 0} \nabla_{\boldsymbol{p}}\boldsymbol{F}(\boldsymbol{p}; \kappa) = \boldsymbol{J}(\boldsymbol{p})$. With some more refined analysis, it is easy to see that the convergence above is uniform over $\boldsymbol{p} \in \mathsf{B}(\boldsymbol{p}_0, \varepsilon)$ for small $\varepsilon$. This proves the lemma. $\qquad\square$

### C.1.3 Proof of Lemma C.6

In this proof, we follow the same notations with the proof of Lemma C.5 as in Section C.1.3.

For any $(\bar{\tau}, \bar{\lambda}, \bar{b}, \kappa) \in \overline{\Omega} \times (0, \kappa_0)$ where $\overline{\Omega} = \mathbb{R}_{\geq 0} \times \mathbb{R}_{\geq 0} \times \mathbb{R}$, we define

$$
\begin{aligned}
f_1(\boldsymbol{p}, \kappa) &= \mathbb{E}\Big[ e'_{\ell_{\bar{b}\kappa + z_\alpha}}(\bar{\tau}\sqrt{\kappa}G + Z; \bar{\lambda}\kappa)^2 \Big], \\
f_2(\boldsymbol{p}, \kappa) &= \mathbb{E}\Big[ e''_{\ell_{\bar{b}\kappa + z_\alpha}}(\bar{\tau}\sqrt{\kappa}G + Z; \bar{\lambda}\kappa) \Big], \\
f_3(\boldsymbol{p}, \kappa) &= \mathbb{E}\Big[ e'_{\ell_{\bar{b}\kappa + z_\alpha}}(\bar{\tau}\sqrt{\kappa}G + Z; \bar{\lambda}\kappa) \Big].
\end{aligned}
\tag{27}
$$

By the definition of $F_1, F_2, F_3$ as in Eq. (23), we have

$$
\begin{aligned}
F_1(\boldsymbol{p}, \kappa) &= \bar{\tau}^2 - \bar{\lambda}^2 f_1(\boldsymbol{p}, \kappa), \\
F_2(\boldsymbol{p}, \kappa) &= \bar{\tau} - \bar{\tau}\bar{\lambda}f_2(\boldsymbol{p}, \kappa), \\
F_3(\boldsymbol{p}, \kappa) &= \kappa^{-1}f_3(\boldsymbol{p}, \kappa).
\end{aligned}
\tag{28}
$$

Then, Lemma C.6 holds as long as we show that there exists continuous functions $\boldsymbol{T}(\boldsymbol{p}) = (T_1(\boldsymbol{p}), T_2(\boldsymbol{p}), T_3(\boldsymbol{p}))$ and $\boldsymbol{g}(\boldsymbol{p}) = (g_1(\boldsymbol{p}), g_2(\boldsymbol{p}), g_3(\boldsymbol{p}))$ such that

$$f_1(\boldsymbol{p}, \kappa) = T_1(\boldsymbol{p}) + o(1), \tag{29}$$

$$\partial_\kappa f_1(\boldsymbol{p}, \kappa) = -\bar{\lambda}^{-2} g_1(\boldsymbol{p}) + o(1), \tag{30}$$

$$f_2(\boldsymbol{p}, \kappa) = T_2(\boldsymbol{p}) + o(1), \tag{31}$$

$$\partial_\kappa f_2(\boldsymbol{p}, \kappa) = -(\bar{\tau}\bar{\lambda})^{-1} g_2(\boldsymbol{p}) + o(1), \tag{32}$$

$$f_3(\boldsymbol{p}, \kappa) = o(1), \tag{33}$$

$$\partial_\kappa f_3(\boldsymbol{p}, \kappa) = T_3(\boldsymbol{p}) + o(1), \tag{34}$$

$$\partial_\kappa^2 f_3(\boldsymbol{p}, \kappa) = g_3(\boldsymbol{p}) + o(1), \tag{35}$$

where the $o(1)$ terms convergence to 0 uniformly over $\boldsymbol{p} \in \mathsf{B}(\boldsymbol{p}_0, \varepsilon)$ as $\kappa \to 0+$. Moreover, we need

$$T_1(\boldsymbol{p}_0) = \bar{\tau}_0^2 / \bar{\lambda}_0^2, \tag{36}$$

$$T_2(\boldsymbol{p}_0) = 1/\bar{\lambda}_0, \tag{37}$$

$$T_3(\boldsymbol{p}_0) = 0. \tag{38}$$

We first prove Eq. (29), (30) and (36). First, we have (c.f. Eq. (26))

$$\lim_{\kappa \to 0+} f_1(\boldsymbol{p}, \kappa) = \lim_{\kappa \to 0+} \mathbb{E}\left[ \frac{1}{\bar{\lambda}^2 \kappa^2} \int_{[\overline{G}_-, \overline{G}_+]} (z - \overline{G})^2 \phi_z(z) \mathrm{d}z + \alpha^2[1 - \Phi_z(\overline{G}_+)] + (1 - \alpha)^2 \Phi_z(\overline{G}_-) \right]$$

$$= \alpha^2(1 - \Phi_z(z_\alpha)) + (1 - \alpha)^2 \Phi_z(z_\alpha) = \alpha(1 - \alpha) = \bar{\tau}_0^2 / \bar{\lambda}_0^2.$$

where the last equality is by the definition in Eq. (17). Further, by smoothness of the density $\phi_z$, and the fact that the neighborhood $\mathsf{B}(\boldsymbol{p}_0, \varepsilon)$ is bounded, this convergence is uniform over $\boldsymbol{p} = (\bar{\tau}, \bar{\lambda}, \bar{b}) \in \mathsf{B}(\boldsymbol{p}_0, \varepsilon)$. This proves Eq. (29) and (36).

Moreover, we have

$$\partial_\kappa f_1(\boldsymbol{p}, \kappa)$$

$$= \mathbb{E}\Bigg[ -\frac{2}{\bar{\lambda}^2 \kappa^3} \int_{[\overline{G}_-, \overline{G}_+]} (z - \overline{G})^2 \phi_z(z) \mathrm{d}z$$

$$+ \frac{1}{\bar{\lambda}^2 \kappa^2} (\overline{G}_+ - \overline{G})^2 \phi_z(\overline{G}_+)(\bar{b} + \alpha\bar{\lambda} - G\bar{\tau}/(2\sqrt{\kappa}))$$

$$- \frac{1}{\bar{\lambda}^2 \kappa^2} (\overline{G}_- - \overline{G})^2 \phi_z(\overline{G}_-)(\bar{b} - (1 - \alpha)\bar{\lambda} - G\bar{\tau}/(2\sqrt{\kappa}))$$

$$- \alpha^2 \phi_z(\overline{G}_+)(\bar{b} + \alpha\bar{\lambda} - G\bar{\tau}/(2\sqrt{\kappa})) + (1 - \alpha)^2 \phi_z(\overline{G}_-)(\bar{b} - (1 - \alpha)\bar{\lambda} - G\bar{\tau}/(2\sqrt{\kappa})) \Bigg]$$

$$= \mathbb{E}\Bigg[ -\frac{2}{\bar{\lambda}^2 \kappa^3} \int_{[\overline{G}_-, \overline{G}_+]} (z - \overline{G})^2 \phi_z(z) \mathrm{d}z \Bigg],$$

where the last inequality is by Stein's identity for $Z \sim \mathcal{N}(0, 1)$ and a consequence of many cancellation happening. So this gives

$$\lim_{\kappa \to 0+} \partial_\kappa f_1(\boldsymbol{p}, \kappa) = -\frac{2}{3\bar{\lambda}^2}\Big[ \alpha^2 - (1 - \alpha)^3 \Big] \phi_z(z_\alpha).$$

Again, by the smoothness of $\phi_z$, and the fact that the neighborhood $\mathsf{B}(\boldsymbol{p}_0, \varepsilon)$ is bounded, this convergence is uniform over $\boldsymbol{p} = (\bar{\tau}, \bar{\lambda}, \bar{b}) \in \mathsf{B}(\boldsymbol{p}_0, \varepsilon)$. This proves Eq. (30). The proof of other equations within (29) to (38) follow from similar continuity arguments. This proves Lemma C.6. □

### C.1.4 Proof of Lemma C.1 and Lemma C.2

We consider the function $\boldsymbol{F}$ defined in (23). First, by Lemma C.6, we have $\boldsymbol{F}(\boldsymbol{p}_0, 0_+) = \boldsymbol{0}$. Further, by Lemma C.5 and C.6, the conditions in the Implicit Function Theorem (Lemma A.1) are satisfied, from which we can conclude that there exists $\kappa_0 > 0$ and a continuously differentiable path $\{\boldsymbol{p}(\kappa) = (\bar{\tau}(\kappa), \bar{\lambda}(\kappa), \bar{b}(\kappa)) : \kappa \in [0, \kappa_0)\} \subset \mathsf{B}(\boldsymbol{p}_0, \varepsilon)$, such that $\boldsymbol{F}(\boldsymbol{p}(\kappa), \kappa) = 0$ for any $\kappa \in [0, \kappa_0)$. Therefore, the set of variables

$$(\tau_\star(\kappa), \lambda_\star(\kappa), b_\star(\kappa)) = \big( \bar{\tau}(\kappa) \cdot \kappa, \ \bar{\lambda}(\kappa) \cdot \kappa, \ z_\alpha + \bar{b} \cdot \kappa \big),$$

is a unique solution to the original system of equations (16) by the equivalence between system (16) and system (22) under this change of variables. This proves Lemma C.1.

In order to prove Lemma C.2 (the local linear expansion), it suffices to prove that $p(\kappa) \to p_0 = (\bar\tau_0, \bar\lambda_0, \bar{b}_0)$. This was already implied by the continuity of $p(\kappa)$ w.r.t. $\kappa$ as stated above. $\qquad\square$

## C.2  Connection between system of equations (16) and a variational problem

Define

$$D(\tau, b, \tau_g, \beta) \equiv \left[ \frac{\beta\tau_g}{2} + \frac{1}{\kappa} \mathbb{E}_{(G,Z)\sim\mathcal{N}(0,1)\times P_z} [e_{\ell_b}(\tau G + Z; \tau_g/\beta)] - \tau\beta \right]. \tag{39}$$

The $D$ defined above is strictly convex-concave as stated in the following lemma.

**Lemma C.7** (Strict convexity-concavity). *Suppose $\kappa \in (0, 1)$. Then for any $(\tau, b, \tau_g, \beta) \in \mathbb{R}_{>0} \times \mathbb{R} \times \mathbb{R}_{>0} \times \mathbb{R}_{>0}$, the function $D$ defined in (39) is strictly convex in $(\tau, b, \tau_g)$ ($\nabla^2_{\tau,b,\tau_g} D \succ \mathbf{0}$), and strictly concave in $\beta$.*

*Proof of Lemma C.7.*  Define

$$E(\tau, b, \tau_g, \beta) \equiv \mathbb{E}_{(G,Z)\sim\mathcal{N}(0,1)\times P_z} [e_{\ell_b}(\tau G + Z; \tau_g/\beta)].$$

We write in short $\partial_x e = \partial_x e_{\ell_b}(\tau G + Z; \tau_g/\beta)$ and $\partial_x^2 e = \partial_x \partial_x e_{\ell_b}(\tau G + Z; \tau_g/\beta)$. Then by Eq. (14), we have

$$\partial_\tau E(\tau, b, \tau_g, \beta) \equiv \mathbb{E}[\partial_x e \cdot G],$$
$$\partial_b E(\tau, b, \tau_g, \beta) \equiv -\mathbb{E}[\partial_x e],$$
$$\partial_{\tau_g} E(\tau, b, \tau_g, \beta) \equiv -\frac{1}{2\beta}\mathbb{E}[(\partial_x e)^2],$$
$$\partial_\beta E(\tau, b, \tau_g, \beta) \equiv \frac{\tau_g}{2\beta^2}\mathbb{E}[(\partial_x e)^2].$$

By Eq. (15), for any $(\tau, b, \tau_g, \beta) \in \mathbb{R}_{>0} \times \mathbb{R} \times \mathbb{R}_{>0} \times \mathbb{R}_{>0}$, we have

$$\partial_\beta^2 E = -\frac{\tau_g}{\beta^3}\mathbb{E}[(\partial_x e)^2] + \frac{\tau_g^2}{\beta^4}\mathbb{E}[(\partial_x e)^2 \partial_x^2 e] = -\frac{\tau_g}{\beta^3}\mathbb{E}[(\partial_x e)^2 \mathbf{1}\{\mathrm{prox}_{\ell_b}(\tau G + Z) \neq b\}] < 0.$$

This gives $\partial_\beta^2 D = \kappa^{-1}\partial_\beta^2 E < 0$, so that $D$ is strictly concave in $\beta$ (for any fixed $(\tau, b, \tau_g)$).

By Eq. (15) again, we have

$$\nabla^2_{(\tau,b,\tau_g)}E = \mathbb{E}\begin{bmatrix} G^2 \cdot \partial_x^2 e & -G \cdot \partial_x^2 e & -\beta^{-1}\partial_x e \cdot G \cdot \partial_x^2 e \\ -G \cdot \partial_x^2 e & \partial_x^2 e & \beta^{-1}\partial_x e \cdot \partial_x^2 e \\ -\beta^{-1}\partial_x e \cdot G \cdot \partial_x^2 e & \beta^{-1}\partial_x e \cdot \partial_x^2 e & \beta^{-2}(\partial_x e)^2 \cdot \partial_x^2 e \end{bmatrix}$$
$$= \frac{\beta}{\tau_g}\mathbb{E}[\mathbf{1}\{\mathrm{prox}_{\ell_b}(\tau G + Z; \tau_g/\beta) \neq b\} \cdot \boldsymbol{u}\boldsymbol{u}^\top].$$

where $\boldsymbol{u} = (G, -1, \beta^{-1}\partial_x e)$. Note that there exists $(G_1, Z_1)$, $(G_2, Z_2)$ and $(G_3, Z_3)$ such that $\mathrm{prox}_{\ell_b}(\tau G_1 + Z_1; \tau_g/\beta), \mathrm{prox}_{\ell_b}(\tau G_2 + Z_2; \tau_g/\beta), \mathrm{prox}_{\ell_b}(\tau G_3 + Z_3; \tau_g/\beta) \neq b$, and

$$\begin{bmatrix} G_1 & -1 & \beta^{-1}\partial_x e_{\ell_b}(\tau G_1 + Z_1; \tau_g/\beta) \\ G_2 & -1 & \beta^{-1}\partial_x e_{\ell_b}(\tau G_2 + Z_2; \tau_g/\beta) \\ G_3 & -1 & \beta^{-1}\partial_x e_{\ell_b}(\tau G_3 + Z_3; \tau_g/\beta) \end{bmatrix}$$

is full rank. By Lemma A.3, we have $\nabla^2_{(\tau,b,\tau_g)}E \succ 0$. Note that $\nabla^2_{(\tau,b,\tau_g)}D = \kappa^{-1}\nabla^2_{(\tau,b,\tau_g)}E \succ 0$, so that $D$ is strictly convex in $(\tau, b, \tau_g)$ (for any fixed $\beta$). This proves the lemma. $\qquad\square$

We now characterize a min-max variational problem associated with the function $D$, and show that it has a unique solution for small $\kappa$, and the solution is related to the solution of the system of equations (16).

**Lemma C.8** (Characterization of variational problem). *Consider the following variational problem in four variables over the function $D$ defined in* (39)*:*

$$\inf_{\tau>0,b\in\mathbb{R},\tau_g>0}\sup_{\beta>0} D(\tau,b,\tau_g,\beta)$$

$$= \inf_{\tau>0,b\in\mathbb{R},\tau_g>0}\sup_{\beta>0}\Big[\frac{\beta\tau_g}{2}+\frac{1}{\kappa}\mathbb{E}_{(G,Z)\sim\mathcal{N}(0,1)\times P_z}[e_{\ell_b}(\tau G+Z;\tau_g/\beta)]-\tau\beta\Big]. \tag{40}$$

*For all sufficiently small $\kappa\in(0,\kappa_0]$, there exists a unique solution $(\widetilde{\tau}_\star,\widetilde{b}_\star,\widetilde{\tau}_{g,\star},\widetilde{\beta}_\star)$ (which depends on $\kappa$) to problem* (40). *This solution is related to the solution $(\tau_\star(\kappa),\lambda_\star(\kappa),b_\star(\kappa))$ of* (16) *as*

$$\widetilde{\tau}_\star = \widetilde{\tau}_{g,\star} = \tau_\star(\kappa),\ \ \widetilde{\beta}_\star = \tau_\star(\kappa)/\lambda_\star(\kappa),\ \ \widetilde{b}_\star = b_\star(\kappa). \tag{41}$$

*Further, for some positive $\varepsilon>0$, for any $b'\in[b_\star-\varepsilon,b_\star+\varepsilon]$, the following variational problem in three variables*

$$\inf_{\tau>0,\tau_g>0}\sup_{\beta>0} D(\tau,b',\tau_g,\beta)$$

$$= \inf_{\tau>0,b\in\mathbb{R},\tau_g>0}\sup_{\beta>0}\Big[\frac{\beta\tau_g}{2}+\frac{1}{\kappa}\mathbb{E}_{(G,Z)\sim\mathcal{N}(0,1)\times P_z}\big[e_{\ell_{b'}}(\tau G+Z;\tau_g/\beta)\big]-\tau\beta\Big] \tag{42}$$

*has a unique solution within $\mathbb{R}^3_{>0}$.*

*Proof of Lemma C.8.* Calculating the derivatives of $D(\tau,b,\tau_g,\beta)$, we get

$$\partial_\tau D(\tau,b,\tau_g,\beta) = \kappa^{-1}\mathbb{E}[Ge'_{\ell_b}(\tau G+Z;\tau_g/\beta)]-\beta,$$

$$\partial_b D(\tau,b,\tau_g,\beta) = -\kappa^{-1}\mathbb{E}[e'_{\ell_b}(\tau G+Z;\tau_g/\beta)],$$

$$\partial_{\tau_g} D(\tau,b,\tau_g,\beta) = \beta/2 - \frac{1}{2\kappa\beta}\mathbb{E}[e'_{\ell_b}(\tau G+Z;\tau_g/\beta)^2],$$

$$\partial_\beta D(\tau,b,\tau_g,\beta) = \tau_g/2 - \tau + \frac{\tau_g}{2\kappa\beta^2}\mathbb{E}[e'_{\ell_b}(\tau G+Z;\tau_g/\beta)^2].$$

By Lemma C.1, there exists $\kappa_0>0$ such that for any $\kappa\in(0,\kappa_0]$, there exists a unique solution $(\tau_\star(\kappa),\lambda_\star(\kappa),b_\star(\kappa))$ of Eq. (16). Plugging in $(\tau,b,\tau_g,\beta)=(\tau_\star(\kappa),b_\star(\kappa),\tau_\star(\kappa),\tau_\star(\kappa)/\lambda_\star(\kappa))$ into the derivatives above and using Eq. (16), we get $\nabla_{(\tau,b,\tau_g,\beta)}D(\tau_\star(\kappa),b_\star(\kappa),\tau_\star(\kappa),\tau_\star(\kappa)/\lambda_\star(\kappa))=0$. This proves that $(\widetilde{\tau}_\star,\widetilde{b}_\star,\widetilde{\tau}_{g,\star},\widetilde{\beta}_\star)=(\tau_\star(\kappa),b_\star(\kappa),\tau_\star(\kappa),\tau_\star(\kappa)/\lambda_\star(\kappa))$ is a stationary point of $D$.

Since $D$ is jointly strictly convex in $(\tau,b,\tau_g)$ and strictly concave in $\beta$ as stated in Lemma C.7, we get

$$\inf_{\tau>0,b\in\mathbb{R},\tau_g>0}\sup_{\beta>0} D(\tau,b,\tau_g,\beta) \le \sup_{\beta>0}D(\widetilde{\tau}_\star,\widetilde{b}_\star,\widetilde{\tau}_{g,\star},\beta)=D(\widetilde{\tau}_\star,\widetilde{b}_\star,\widetilde{\tau}_{g,\star},\widetilde{\beta}_\star),$$

$$\inf_{\tau>0,b\in\mathbb{R},\tau_g>0}\sup_{\beta>0} D(\tau,b,\tau_g,\beta) \ge \inf_{\tau>0,b\in\mathbb{R},\tau_g>0}D(\tau,b,\tau_g,\widetilde{\beta}_\star)=D(\widetilde{\tau}_\star,\widetilde{b}_\star,\widetilde{\tau}_{g,\star},\widetilde{\beta}_\star).$$

This proves that $(\widetilde{\tau}_\star,\widetilde{b}_\star,\widetilde{\tau}_{g,\star},\widetilde{\beta}_\star)$ is a solution of the variational problem (40). By the strict convexity-concavity property of $D$ again, the solution of the variational problem (40) is unique. Finally, the existence and uniqueness of the solution of $\inf_{\tau>0,\tau_g>0}\sup_{\beta>0} D(\tau,b',\tau_g,\beta)$ for $b'\in[b_\star-\varepsilon,b_\star+\varepsilon]$ follows from similar arguments. $\square$

### C.3   Proof of Theorem C.1

**Preliminary: the asymptotic limit fixed $b$ via CGMT**   For any convex function $\ell:\mathbb{R}\to\mathbb{R}$, we define notation

$$\ell'_+(v) \equiv \sup_{s\in\partial\ell(v)}|s|.$$

For $\tau>0$, we define (with some abuse of notation)

$$D(\tau) \equiv \inf_{\tau_g>0}\sup_{\beta>0}\Big[\frac{\beta\tau_g}{2}+\frac{1}{\kappa}\mathbb{E}_{(G,Z)\sim\mathcal{N}(0,1)\times P_z}[e_\ell(\tau G+Z;\tau_g/\beta)]-\tau\beta\Big]. \tag{43}$$

The following proposition is by [62, Theorem 4.1], which uses the Convex Gaussian Comparison Theorem (CGMT).

**Proposition C.1** (A simplification of Theorem 4.1 in [62] up to model rescaling)**.** *Let $\ell$ be a closed proper convex function and $P_z$ be a distribution on the real line satisfying*

- $\mathbb{E}_{(G,Z)\sim\mathcal{N}(0,1)\times P_z}[|\ell'_+(cG+Z)|^2] < \infty$*, for all $c \in \mathbb{R}$;*

- $\sup_{v\in\mathbb{R}}|\ell'_+(v)| < \infty$*.*

*Further assume that the set $\arg\min_\tau D(\tau)$ is bounded for the function $D$ defined in (43). Then $D$ has a unique minimizer $\tau_\star > 0$. Moreover, in the limit $n, d \to \infty$ and $d/n \to \kappa$, we have*

$$\min_{\mathbf{w}} \frac{1}{n}\sum_{i=1}^{n}\ell(y_i - \langle\mathbf{x}_i,\mathbf{w}\rangle) \xrightarrow{p} \min_\tau D(\tau).$$

*Furthermore, for any $\varepsilon > 0$, defining $S_\varepsilon \equiv \{\mathbf{w} : |\|\mathbf{w}-\mathbf{w}_\star\|_2^2 - \tau_\star^2| \le \varepsilon\}$, there exists $\delta > 0$ such that*

$$\min_{\mathbf{w}\in S_\varepsilon^c} \frac{1}{n}\sum_{i=1}^{n}\ell(y_i - \langle\mathbf{x}_i,\mathbf{w}\rangle) \xrightarrow{p} \min_\tau D(\tau) + \delta.$$

*As a consequence, for any empirical risk minimizer $\widehat{\mathbf{w}}$ satisfying*

$$\widehat{\mathbf{w}} \in \arg\min_{\mathbf{w}} \frac{1}{n}\sum_{i=1}^{n}\ell(y_i - \langle\mathbf{x}_i,\mathbf{w}\rangle),$$

*we have*

$$\|\widehat{\mathbf{w}} - \mathbf{w}_\star\|_2^2 \xrightarrow{p} \tau_\star^2.$$

We are now ready to prove Theorem C.1.

*Proof of Theorem C.1.* We define (with some abuse of notation)

$$D(\tau, b) \equiv \inf_{\tau_g>0}\sup_{\beta>0}\left[\frac{\beta\tau_g}{2} + \frac{1}{\kappa}\mathbb{E}_{(G,Z)\sim\mathcal{N}(0,1)\times P_z}\left[e_{\ell_b^\alpha}(\tau G + Z; \tau_g/\beta)\right] - \tau\beta\right]. \qquad (44)$$

**Step 1. Show that $\widehat{b} \xrightarrow{p} b_\star$.** For any fixed $b \in \mathbb{R}$, define the associated minimum empirical risk (over $\mathbf{w} \in \mathbb{R}^d$) as

$$L_n(b) \equiv \min_{\mathbf{w}} \widehat{R}_n(\mathbf{w}, b).$$

Notice that $\widehat{b} = \arg\min_{b\in\mathbb{R}} L_n(b)$. Let $(\tau_\star, \kappa_\star, b_\star)$ be defined as in Lemma C.1 (as well as Lemma C.8). By Lemma C.8, there exists some $\varepsilon > 0$ such that for any fixed $b \in [b_\star - \varepsilon, b_\star + \varepsilon]$, we have $\arg\min_\tau D(\tau)$ is a singleton. Therefore the conditions of Proposition C.1 is satisfied, from which we conclude that

$$L_n(b) \xrightarrow{p} \min_\tau D(\tau, b).$$

Now, observe that $\min_\tau D(\tau, b) = \min_{\tau,\tau_g}\max_\beta D(\tau, b, \tau_g, \beta)$ is strictly convex in $b$ (this is because $D(\tau, b, \tau_g, \beta)$ has a positive definite Hessian w.r.t. $(\tau, b, \tau_g)$ at any $(\tau, b, \tau_g, \beta)$ by Lemma C.7). Then for any $\varepsilon > 0$, there exists $\delta > 0$ such that

$$\min_\tau D(\tau, b_\star + \varepsilon) \ge \min_\tau D(\tau, b_\star) + \delta, \qquad \min_\tau D(\tau, b_\star - \varepsilon) \ge \min_\tau D(\tau, b_\star) + \delta.$$

As a consequence, with probability going to 1, we have the event

$$\{L_n(b_\star + \varepsilon) > L_n(b_\star) + \delta/2, \quad L_n(b_\star - \varepsilon) > L_n(b_\star) + \delta/2\}.$$

Furthermore, since $L_n(b)$ is a convex function in $b$, this implies that, with probability going to 1, we have $|\widehat{b} - b_\star| \le \varepsilon$. Note that this is for any $\varepsilon > 0$. This proves that $\widehat{b} \xrightarrow{p} b_\star$.

**Step 2. Show that $\|\widehat{\mathbf{w}} - \mathbf{w}_\star\|_2^2 \xrightarrow{p} \tau_\star^2$.** By Proposition C.1, for any $\varepsilon > 0$, there exists $\delta > 0$ such that

$$\min_{\mathbf{w}\in S_\varepsilon^c} \widehat{R}_n(\mathbf{w}, b_\star) \xrightarrow{p} \min_\tau D(\tau, b_\star) + \delta.$$

where $S_\varepsilon \equiv \{\mathbf{w} : |\|\mathbf{w}-\mathbf{w}_\star\|_2^2 - \tau_\star^2| \le \varepsilon\}$.

Furthermore, note that $\ell^\alpha(t) = -(1-\alpha)t\mathbf{1}\{t \le 0\} + \alpha t\mathbf{1}\{t > 0\}$ is a 1-Lipschitz function in $t$, this gives

$$\sup_{\mathbf{w}} \left| \widehat{R}_n(\mathbf{w}, b_1) - \widehat{R}_n(\mathbf{w}, b_2) \right| \le |b_1 - b_2|$$

As a consequence, we have

$$\min_{\mathbf{w} \in S_\varepsilon^c} \widehat{R}_n(\mathbf{w}, \widehat{b}) \ge \min_{\mathbf{w} \in S_\varepsilon^c} \widehat{R}_n(\mathbf{w}, b_\star) - |\widehat{b} - b_\star| \xrightarrow{p} \min_\tau D(\tau, b_\star) + \delta.$$

In the mean time, by Proposition C.1, we have

$$\min_{\mathbf{w}} \widehat{R}_n(\mathbf{w}, \widehat{b}) \le \min_{\mathbf{w}} \widehat{R}_n(\mathbf{w}, b_\star) + |\widehat{b} - b_\star| \xrightarrow{p} \min_\tau D(\tau, b_\star).$$

This implies that, with probability approaching 1, we have

$$\min_{\mathbf{w} \in S_\varepsilon^c} \widehat{R}_n(\mathbf{w}, b_\star) \ge \min_\tau D(\tau, b_\star) + 2\delta/3 \quad \text{and} \quad \min_{\mathbf{w}} \widehat{R}_n(\mathbf{w}, \widehat{b}) \le \min_\tau D(\tau, b_\star) + \delta/3.$$

On this event we have $\widehat{\mathbf{w}} \in S_\varepsilon$. Note that this is for any $\varepsilon > 0$. This proves that $\|\widehat{\mathbf{w}} - \mathbf{w}_\star\|_2^2 \xrightarrow{p} \tau_\star^2$. $\quad\square$

### C.4 Proof of Lemma C.3 and Lemma C.4

Recall that

$$\text{Coverage}(\widehat{f}) = \mathbb{P}_{(\mathbf{x},y)}\left( y \le \widehat{\mathbf{w}}^\top \mathbf{x} + \widehat{b} \right) = \mathbb{E}_{G \sim \mathsf{N}(0,1)}\left[ \Phi_z\left( \|\widehat{\mathbf{w}} - \mathbf{w}_\star\|_2 G + \widehat{b} \right) \right].$$

Eq. (20) is simply by the fact that $T(\tau, b; G) \equiv \Phi_z(\tau G + b)$ is a continuous function in $(\tau, b)$, by Theorem C.1, and by the dominant convergence theorem. This proves Lemma C.3.

Furthermore, by Taylor expansion, we have

$$\text{Coverage}_{\alpha,\kappa} = \mathbb{E}[\Phi_z(\tau_\star(\kappa)G + b_\star(\kappa))]$$

$$= \Phi_z(z_\alpha) + \phi_z(z_\alpha)\mathbb{E}[(\tau_\star(\kappa)G + b_\star(\kappa) - z_\alpha)] + \frac{1}{2}\phi_z'(z_\alpha)\mathbb{E}[(\tau_\star(\kappa)G + b_\star(\kappa) - z_\alpha)^2]$$

$$+ \frac{1}{6}\mathbb{E}[\phi_z''(\xi)(\tau_\star(\kappa)G + b_\star(\kappa) - z_\alpha)^3]$$

$$= \alpha + \phi_z(z_\alpha)(b_\star(\kappa) - z_\alpha) + \frac{1}{2}\phi_z'(z_\alpha)\tau_\star^2(\kappa) + o(\kappa)$$

$$= \alpha + \left( \phi_z(z_\alpha)\bar{b}_0 + \frac{1}{2}\phi_z'(z_\alpha)\bar{\tau}_0^2 \right)\kappa + o(\kappa),$$

where the last equality is by Lemma C.2 and by the boundedness of $\phi_z''$. This proves Eq. (21) and thus Lemma C.4.

## D   Extension to over-parametrized learning

In this section we provide a variant of Theorem 1 in the over-parametrized case, i.e. when $d \ge n$, so that the learned quantile functions have the capacity to interpolate the entire training dataset. We still assume that the data are generated from the linear model (4). For notational simplicity, throughout this section we let $\boldsymbol{\theta} := [\mathbf{w}^\top, b]^\top \in \mathbb{R}^{d+1}$ denote the concatenation of $\mathbf{w}$ and $b$, and let $\widehat{R}_n(\boldsymbol{\theta})$ denote the empirical risk (6). We also let $\widetilde{\mathbf{x}} = [\mathbf{x}^\top, 1]^\top \in \mathbb{R}^{d+1}$ denote the augmented feature so that $\boldsymbol{\theta}^\top\widetilde{\mathbf{x}} = \mathbf{w}^\top\mathbf{x} + b$. We let $\widetilde{\mathbf{X}} \in \mathbb{R}^{n \times (d+1)}$ denote the augmented input matrix and $\mathbf{z} \in \mathbb{R}^n$ denote the noise vector.

In the over-parametrized case, the ERM is no longer well-defined as there are multiple interpolating solutions. We consider instead the quantile functions obtained on the gradient descent path on the empirical risk $\widehat{R}_n$. More precisely, we consider the vanilla (sub-)gradient descent algorithm: Initialize $\boldsymbol{\theta}_1 = \mathbf{0}$, and iterate for all $t \ge 1$

$$\boldsymbol{\theta}_{t+1} = \boldsymbol{\theta}_t - \eta_t \mathbf{g}_t, \tag{45}$$

where $\mathbf{g}_t \in \partial\widehat{R}_n(\boldsymbol{\theta}_t)$ is any sub-gradient of the empirical risk $\widehat{R}_n$ (6) at $\boldsymbol{\theta}_t$.

**Theorem D.1** (Quantile regression under over-parametrization). *Suppose the data is generated from the Gaussian linear model* (4) *with* $\|\mathbf{w}\|_2 = R$, *and the nominal quantile level* $\alpha \in (0.5, 1)$. *Further assume the noise distribution* $P_z$ *is symmetric about* $0$ *and* $\sigma^2$-*sub-Gaussian. Then, there exists an absolute constant* $C_0 > 0$ *such that if* $n \geq C_0(d + \log(1/\delta))$, *the following holds.*

*Let* $\boldsymbol{\theta}_t$ *be the iterates of the sub-gradient descent algorithm* (45) *with step-size* $\eta_t := \beta/\sqrt{t}$ *for any* $\beta > 0$, *and let* $\boldsymbol{\theta}_\infty \in \mathbb{R}^{d+1}$ *denote any limit point of* $\{\boldsymbol{\theta}_t\}_{t \geq 1}$, *then we have*

(a) *$\boldsymbol{\theta}_\infty$ is the minimum $\ell_2$-norm interpolator of the training data, i.e.*

$$\boldsymbol{\theta}_\infty = \arg\min_{\boldsymbol{\theta} \in \mathbb{R}^d} \left\{ \|\boldsymbol{\theta}\|_2 : \widetilde{\mathbf{X}}\boldsymbol{\theta} = \mathbf{y} \right\}.$$

(b) *With probability at least $1 - \delta$ (over the training data), the coverage of the limiting quantile function $\widehat{f}_\infty := \boldsymbol{\theta}_\infty^\top \widetilde{\mathbf{x}} = \mathbf{w}_\infty^\top \mathbf{x} + b_\infty$ concentrates around $0.5$:*

$$\left| \mathrm{Coverage}(\widehat{f}_\infty) - 0.5 \right| \leq C(R + \sigma) \cdot \sqrt{\frac{\log(1/\delta)}{d}},$$

*where $C > 0$ is a constant that only depends on $\sup_{t \in \mathbb{R}} |\phi_z(t)|$.*

**Implications** Theorem D.1 shows that a severe under-coverage bias in the over-parametrized case: The coverage of the limiting quantile function (of the gradient descent path) is $0.5 \pm \widetilde{O}(1/\sqrt{d})$, *regardless of the nominal quantile level* $\alpha \in (0.5, 1)$. Therefore $\widehat{f}_\infty$ under-covers by $\alpha - 0.5 = \Theta(1)$, and this under-coverage bias does not diminish as we increase $n, d$.

The proof of Theorem D.1 is established in the following two subsections.

## D.1 Proof of Part (a)

We begin by observing that the sub-gradients of the quantile risk (6) takes the form

$$\mathbf{g}_t = \frac{1}{n} \sum_{i=1}^n (\ell^\alpha)'(y_i - \boldsymbol{\theta}_t^\top \widetilde{\mathbf{x}}_i) \cdot \widetilde{\mathbf{x}}_i \in \mathrm{span}\{\widetilde{\mathbf{x}}_1, \ldots, \widetilde{\mathbf{x}}_n\}, \tag{46}$$

where $(\ell^\alpha)'(t)$ is the sub-gradient of $\ell^\alpha$, which takes value $-(1 - \alpha)$ at $t < 0$, $\alpha$ at $t > 0$, and any value within $[-(1 - \alpha), \alpha]$ at $t = 0$. As we initialized at $\boldsymbol{\theta}_1 = \mathbf{0}$, this implies that

$$\boldsymbol{\theta}_t \in \mathrm{span}\{\widetilde{\mathbf{x}}_1, \ldots, \widetilde{\mathbf{x}}_n\}$$

for all $t \geq 1$. Also, by (46) we have $\|\mathbf{g}_t\|_2 \leq M := \max_{i \in [n]} \|\widetilde{\mathbf{x}}_i\|_2$, since $|(\ell^\alpha)'| \leq \max\{\alpha, 1 - \alpha\} \leq 1$.

Also, let $\boldsymbol{\theta}_{\ell_2}$ denote the minimum $\ell_2$-norm interpolator of the dataset:

$$\boldsymbol{\theta}_{\ell_2} := \arg\min_{\boldsymbol{\theta} \in \mathbb{R}^d} \left\{ \|\boldsymbol{\theta}\|_2 : \widetilde{\mathbf{X}}\boldsymbol{\theta} = \mathbf{y} \right\} = \widetilde{\mathbf{X}}^\dagger \mathbf{y} = \widetilde{\mathbf{X}}^\top (\widetilde{\mathbf{X}}\widetilde{\mathbf{X}}^\top)^{-1} \mathbf{y}. \tag{47}$$

This $\boldsymbol{\theta}_{\ell_2}$ exists whenever $d + 1 \geq n$ (so that $\widetilde{\mathbf{x}}_i \in \mathbb{R}^{d+1}$ are linearly independent with probability one and thus $\widetilde{\mathbf{X}}\widetilde{\mathbf{X}}^\top \in \mathbb{R}^{n \times n}$ is invertible). It further satisfies

- $\widehat{R}_n(\boldsymbol{\theta}_{\ell_2}) = 0$ (since $\boldsymbol{\theta}_{\ell_2}^\top \widetilde{\mathbf{x}}_i = y_i$). Therefore $\boldsymbol{\theta}_{\ell_2}$ is a minimizer of $\widehat{R}_n$ since $\widehat{R}_n \geq 0$.
- $\boldsymbol{\theta}_{\ell_2} \in \mathrm{span}\{\widetilde{\mathbf{x}}_1, \ldots, \widetilde{\mathbf{x}}_n\}$.
- $\boldsymbol{\theta}_{\ell_2}$ is the only point within $\mathrm{span}\{\widetilde{\mathbf{x}}_1, \ldots, \widetilde{\mathbf{x}}_n\}$ that satisfies $\widehat{R}_n(\boldsymbol{\theta}_{\ell_2}) = 0$, as any such point $\boldsymbol{\theta} \in \mathbb{R}^{d+1}$ must satisfy $\widetilde{\mathbf{X}}\boldsymbol{\theta} = \mathbf{y}$, and there is only one such point in the span because of the linear independence of $\{\widetilde{\mathbf{x}}_i\}_{i=1}^n$.

We now use the following lemma on the last-iterate convergence of sub-gradient descent, adapted from [48, Corollary 3]:

**Lemma D.1** (Last-iterate convergence of sub-gradient descent). *Suppose $F : \mathbb{R}^D \to \mathbb{R}$ is a convex function with bounded sub-gradients: $\|\mathbf{g}\|_2 \leq M$ for all $\mathbf{g} \in \partial F(\boldsymbol{\theta})$ and any $\boldsymbol{\theta} \in \mathbb{R}^D$. Let $\boldsymbol{\theta}_\star \in \mathbb{R}^D$ be any minimizer of $F$ with $F_\star = F(\boldsymbol{\theta}_\star) > -\infty$. Consider the sub-gradient descent algorithm*

$$\boldsymbol{\theta}_{t+1} = \boldsymbol{\theta}_t - \eta_t \mathbf{g}_t,$$

*where $\mathbf{g}_t \in \partial F(\boldsymbol{\theta}_t)$, and $\eta_t = \beta/\sqrt{t}$ for some $\beta > 0$. Then, we have for all $T \geq 3$ that*

$$F(\boldsymbol{\theta}_T) - F_\star \leq \frac{\|\boldsymbol{\theta}_1 - \boldsymbol{\theta}_\star\|_2^2 + 4M^2\beta^2 \log T}{2\beta\sqrt{T}}.$$

Applying Lemma D.1 with on the quantile risk $\widehat{R}_n$ the associated minimizer $\boldsymbol{\theta}_{\ell_2}$, we get that (for $T \geq 3$)

$$\widehat{R}_n(\boldsymbol{\theta}_T) \leq \frac{\|\boldsymbol{\theta}_{\ell_2}\|_2^2 + 4M^2\beta^2 \log T}{2\beta\sqrt{T}}.$$

This implies that $\widehat{R}_n(\boldsymbol{\theta}_T) \to 0$ as $T \to \infty$.

The above implies that any limit point $\boldsymbol{\theta}_\infty$ of the sequence $\{\boldsymbol{\theta}_t\}_{t \geq 1}$ must satisfy

- $\widehat{R}_n(\boldsymbol{\theta}_\infty) = 0$, by continuity of $\widehat{R}_n$;
- $\boldsymbol{\theta}_\infty \in \mathrm{span}(\widetilde{\mathbf{x}}_1, \ldots, \widetilde{\mathbf{x}}_n)$, by the closedness of the span.

Combined with the above assertions on $\boldsymbol{\theta}_{\ell_2}$, this shows that $\boldsymbol{\theta}_\infty = \boldsymbol{\theta}_{\ell_2}$, establishing part (a) of the theorem. $\qquad\square$

### D.2 Proof of part (b)

We first establish a covariance lower bound useful for the subsequent analyses. As $\mathbf{x}_i \sim \mathsf{N}(\mathbf{0}, \mathbf{I}_d)$, the input matrix $\mathbf{X} \in \mathbb{R}^{n \times d}$ has i.i.d. $\mathsf{N}(0,1)$ entries, and thus $\mathbf{X}$'s columns are also i.i.d. $\mathsf{N}(\mathbf{0}, \mathbf{I}_n)$. By standard sub-Gaussian covariance concentration, we have with probability at least $1 - \delta$ that

$$\left\|\frac{1}{d}\mathbf{X}\mathbf{X}^\top - \mathbf{I}_n\right\|_{\mathrm{op}} \leq C\left(\sqrt{\frac{n + \log(1/\delta)}{d}} + \frac{n + \log(1/\delta)}{d}\right)$$

for some absolute constant $C > 0$ (this can be found in e.g. [65, Example 4.7.3]). In particular, we have $\left\|\mathbf{X}\mathbf{X}^\top/d - \mathbf{I}_n\right\|_{\mathrm{op}} \leq 1/4$ provided $d \geq C(n + \log(1/\delta))$. On this event, we have

$$\mathbf{X}\mathbf{X}^\top \succeq \frac{3d}{4}\mathbf{I}_n.$$

We will apply a small variant of this result: as long as $d - 1 \geq C(n + \log(1/\delta))$, we also have for any fixed matrix $\mathbf{V}_\star \in \mathbb{R}^{d \times (d-1)}$ with orthogonal columns that

$$\mathbf{X}\mathbf{V}_\star\mathbf{V}_\star^\top\mathbf{X}^\top \succeq \frac{3(d-1)}{4}\mathbf{I}_n \succeq \frac{d}{2}\mathbf{I}_n. \tag{48}$$

**Bounding $|b_\infty|$** By (47), we have

$$\begin{bmatrix} \mathbf{w}_\infty \\ b_\infty \end{bmatrix} = \boldsymbol{\theta}_\infty = \boldsymbol{\theta}_{\ell_2} = \widetilde{\mathbf{X}}^\top(\widetilde{\mathbf{X}}\widetilde{\mathbf{X}}^\top)^{-1}\mathbf{y} = \widetilde{\mathbf{X}}^\top(\widetilde{\mathbf{X}}\widetilde{\mathbf{X}}^\top)^{-1}(\mathbf{X}\mathbf{w}_\star + \mathbf{z})$$

$$= \begin{bmatrix} \mathbf{X}^\top \\ \mathbf{1}_n^\top \end{bmatrix} (\widetilde{\mathbf{X}}\widetilde{\mathbf{X}}^\top)^{-1}(\mathbf{X}\mathbf{w}_\star + \mathbf{z}).$$

Therefore

$$b_\infty = \mathbf{1}_n^\top(\widetilde{\mathbf{X}}\widetilde{\mathbf{X}}^\top)^{-1}(\mathbf{X}\mathbf{w}_\star + \mathbf{z}) = \underbrace{\mathbf{1}_n^\top\left(\mathbf{X}\mathbf{X}^\top + \mathbf{1}_n\mathbf{1}_n^\top\right)^{-1}\mathbf{X}\mathbf{w}_\star}_{\text{I}} + \underbrace{\mathbf{1}_n^\top\left(\mathbf{X}\mathbf{X}^\top + \mathbf{1}_n\mathbf{1}_n^\top\right)^{-1}\mathbf{z}}_{\text{II}}.$$

We now bound terms I and II separately.

For term I, let us assume for the moment that $\|\mathbf{w}_\star\|_2 = 1$. Let $\mathbf{V}_\star \in \mathbb{R}^{d \times d-1}$ denote the orthogonal complement to the matrix $\mathbf{w}_\star$ (i.e. so that $[\mathbf{w}_\star, \mathbf{V}_\star] \in \mathbb{R}^{d \times d}$ is an orthogonal matrix). We have

$$\mathrm{I} = \mathbf{1}_n^\top \left(\mathbf{X}\mathbf{V}_\star\mathbf{V}_\star^\top\mathbf{X}^\top + \mathbf{X}\mathbf{w}_\star\mathbf{w}_\star^\top\mathbf{X}^\top + \mathbf{1}_n\mathbf{1}_n^\top\right)^{-1}\mathbf{X}\mathbf{w}_\star.$$

As $\mathbf{X}\mathbf{V}_\star\mathbf{V}_\star^\top\mathbf{X}^\top$ is an positive definite matrix with probability one whenever $d - 1 \geq n$, applying Lemma A.2 twice, we get

$$|\mathrm{I}| \leq \left|\mathbf{1}_n^\top\left(\mathbf{X}\mathbf{V}_\star\mathbf{V}_\star^\top\mathbf{X}^\top + \mathbf{1}_n\mathbf{1}_n^\top\right)^{-1}\mathbf{X}\mathbf{w}_\star\right| \leq \left|\mathbf{1}_n^\top\left(\mathbf{X}\mathbf{V}_\star\mathbf{V}_\star^\top\mathbf{X}^\top\right)^{-1}\mathbf{X}\mathbf{w}_\star\right|.$$

Now, notice that $\mathbf{X}\mathbf{V}_\star \in \mathbb{R}^{n \times d-1}$ and $\mathbf{X}\mathbf{w}_\star \in \mathbb{R}^n$ have i.i.d. $\mathsf{N}(0,1)$ entries and are independent of each other. Further, $\mathbf{X}\mathbf{w}_\star \sim \mathsf{N}(\mathbf{0}, \mathbf{I}_n)$, and thus the random variable $\mathbf{1}_n^\top\left(\mathbf{X}\mathbf{V}_\star\mathbf{V}_\star^\top\mathbf{X}^\top\right)^{-1}\mathbf{X}\mathbf{w}_\star$ (conditional on $\mathbf{X}\mathbf{V}_\star$) is $\|\mathbf{v}_\mathrm{I}\|_2^2$-sub-Gaussian (due to the independence between $\mathbf{X}\mathbf{V}_\star$ and $\mathbf{X}\mathbf{w}_\star$), where

$$\|\mathbf{v}_\mathrm{I}\|_2^2 = \mathbf{1}_n^\top\left(\mathbf{X}\mathbf{V}_\star\mathbf{V}_\star^\top\mathbf{X}^\top\right)^{-2}\mathbf{1}_n \leq \frac{4}{d^2}\|\mathbf{1}_n\|_2^2 = \frac{4n}{d^2},$$

where the inequality used the covariance lower bound (48). This shows that

$$|\mathrm{I}| \leq C\sqrt{4n/d^2 \cdot \log(1/\delta)} \leq C\sqrt{\log(1/\delta)/d}$$

with probability at least $1 - \delta$, where the last step used $n \leq d$. It is straightforward to see that, for general $\|\mathbf{w}_\star\|_2 = R$, we have

$$|\mathrm{I}| \leq CR\sqrt{4n/d^2 \cdot \log(1/\delta)} \leq CR\sqrt{\log(1/\delta)/d}. \tag{49}$$

For term II, As $\mathbf{X}$ and $\mathbf{z}$ are independent, the random variable $\mathrm{II} = \mathbf{1}_n^\top(\mathbf{X}\mathbf{X}^\top + \mathbf{1}_n\mathbf{1}_n^\top)^{-1}\mathbf{z}$ (conditional on $\mathbf{X}$) is $\|\mathbf{v}_\mathrm{II}\|_2^2\,\sigma^2$-sub-Gaussian, where

$$\|\mathbf{v}_\mathrm{II}\|_2^2 = \mathbf{1}_n^\top(\mathbf{X}\mathbf{X}^\top + \mathbf{1}_n\mathbf{1}_n^\top)^{-2}\mathbf{1}_n \leq \frac{4}{d^2}\|\mathbf{1}_n\|_2^2 = \frac{4n}{d^2} \leq \frac{4}{d}.$$

Similar as above, we have with probability at least $1 - \delta$ that

$$|\mathrm{II}| \leq C\sigma\sqrt{\log(1/\delta)/d}. \tag{50}$$

Combining (49) and (50), we get with probability at least $1 - \delta$ that (rescaling $3\delta \to \delta$)

$$|b_\infty| \leq C(R + \sigma)\sqrt{\log(1/\delta)/d}. \tag{51}$$

**Bounding the coverage bias**  We now translate the bound on $|b_\infty|$ to a bound on the coverage error $\left|\mathrm{Coverage}(\widehat{f}_\infty) - 0.5\right|$. First, note that by symmetry of the distribution of $(\mathbf{w}_\infty - \mathbf{w}_\star)^\top\mathbf{x}$ and the fact that $\Phi_z(t) + \Phi_z(-t) = 1$ (due to the symmetry of $P_z$), we have

$$\mathbb{E}\left[\Phi_z\left((\mathbf{w}_\infty - \mathbf{w}_\star)^\top\mathbf{x}\right)\right] = \mathbb{E}\left[\frac{1}{2}\left(\Phi_z\left((\mathbf{w}_\infty - \mathbf{w}_\star)^\top\mathbf{x}\right) + \Phi_z\left(-(\mathbf{w}_\infty - \mathbf{w}_\star)^\top\mathbf{x}\right)\right)\right] = 0.5.$$

Therefore we have

$$\left|\mathrm{Coverage}(\widehat{f}_\infty) - 0.5\right| = \left|\mathbb{E}\left[\Phi_z\left((\mathbf{w}_\infty - \mathbf{w}_\star)^\top\mathbf{x} + b_\infty\right) - \Phi_z\left((\mathbf{w}_\infty - \mathbf{w}_\star)^\top\mathbf{x}\right)\right]\right|$$
$$\leq \sup_{t \in \mathbb{R}}|\phi_z(t)| \cdot |b_\infty|$$
$$\leq C\sup_{t \in \mathbb{R}}|\phi_z(t)| \cdot |b_\infty| \leq C\sup_{t \in \mathbb{R}}|\phi_z(t)| \cdot (R + \sigma)\sqrt{\log(1/\delta)/d}.$$

Notably the bound is also upper bounded by $C\sup_{t \in \mathbb{R}}|\phi_z(t)| \cdot (R + \sigma)\sqrt{\log(1/\delta)/n}$ as we assumed $d \geq n$. This proves part (b) of the theorem. $\qquad\square$

# E  Proofs for Section 4

## E.1  Proof of Corollary 2

First, part (a) is a direct consequence of Lemma C.2 which was established within the proof of Theorem 1.

We now prove part (b). We first show that $\bar{b}_0 < 0$ for $P_z$ being any Gaussian distribution. We first observe that to determine the sign of $\bar{b}_0$, it suffices to consider the standard Gaussian: The value of $\bar{b}_0$ does not depend on the location parameter (since $\phi_z$ and $z_\alpha$ shifts together with a location shift). Also, scalings won't change the sign of $\bar{b}_0$ (although it scales the numerator and the denominator by a different amount).

We next calculate $\bar{b}_0$ for $P_z = \mathsf{N}(0,1)$. We have $\phi'_z(z_\alpha) = -z_\alpha \phi_z(z_\alpha)$ for $\phi_z(t) = \exp(-t^2/2)/\sqrt{2\pi}$. Therefore the numerator of $\bar{b}_0$ is

$$-\alpha(1-\alpha)\phi'_z(z_\alpha) - (2\alpha-1)\phi^2_z(z_\alpha) = (\alpha(1-\alpha)z_\alpha - (2\alpha-1)\phi_z(z_\alpha))\phi_z(z_\alpha).$$

Consider the change of variable $t := z_\alpha$ so that $\alpha = \Phi_z(t)$. To show the above quantity is negative, it suffices to show that

$$\Phi(t)(1-\Phi(t))t - (2\Phi(t)-1)\phi(t) < 0$$

$$\Longleftrightarrow \underbrace{\frac{t(1-\Phi(t))}{\phi(t)} - 2 + \frac{1}{\Phi(t)}}_{:=F(t)} < 0$$

for all $t > 0$, where $\Phi(t) = \Phi_z(t)$ is shorthand for the standard Gaussian CDF. To show this, we first observe that $F(0) = -2 + 1/\Phi(0) = 0$, and further

$$F'(t) = \frac{(1+t^2)(1-\Phi(t))}{\phi(t)} - t - \frac{\phi(t)}{\Phi(t)^2}.$$

We can numerically check that $F'(t) < -0.03$ for $t \in [0,1]$, within which range we have $F(t) < -0.03t < 0$. On the other hand, using the Gaussian CDF approximation bound

$$1 - \frac{1}{t^2} \le \frac{t(1-\Phi(t))}{\phi(t)} \le 1 - \frac{1}{t^2} + \frac{3}{t^4} \quad \text{for all } t > 0,$$

we have

$$F(t) \le 1 - \frac{1}{t^2} + \frac{3}{t^4} - 2 + \frac{1}{1 - (t^{-1} - t^{-3})\phi(t)}$$

$$\overset{(i)}{\le} -\frac{1}{t^2} + \frac{3}{t^4} + 2(t^{-1} - t^{-3})\phi(t) \le \frac{3 + 2t^3\phi(t) - t^2}{t^4} \overset{(ii)}{<} 0,$$

where (i) happens when $(t^{-1} - t^{-3})\phi(t) < 1/2$, which happens for all $t \ge 1$, and (ii) happens when $t \ge 2$. This shows that $F(t) < 0$ for $t \ge 2$. For $t \in [1,2]$, one can check numerically that $F(t) < -0.1 < 0$. This shows $F(t) < 0$ for all $t > 0$, which establishes $\bar{b}_0 < 0$ for $P_z = \mathsf{N}(0,1)$, showing the first claim in part (b).

Next, for any $\alpha \in (0.5, 1)$, we show that there exists a noise distributions $\widetilde{P}_z$ for which $\bar{b}_0 > 0$. Indeed, simply take any smooth density $\phi_z$ (such as standard Gaussian density), and modify $\phi_z$ locally around $z_\alpha$ into some new smooth density $\widetilde{\phi}_z$ such that both the new $\alpha$-quantile $\widetilde{z}_\alpha \approx z_\alpha$ and $\widetilde{\phi}_z(\widetilde{z}_\alpha) \approx \phi_z(z_\alpha)$ (with arbitrarily small differences), but $\widetilde{\phi}'_z(\widetilde{z}_\alpha) < 0$ is negative with a high magnitude $|\widetilde{\phi}'_z(\widetilde{z}_\alpha)|$. Taking this magnitude high enough, we can always make $-\alpha(1-\alpha)\widetilde{\phi}'_z(\widetilde{z}_\alpha) - (2\alpha-1)\widetilde{\phi}_z(\widetilde{z}_\alpha)^2 > 0$, which gives $\bar{b}_0 > 0$ for the noise distribution $\widetilde{P}_z$ defined by the density $\widetilde{\phi}_z$. This shows the second claim in part (b).

## E.2 Proof of Theorem 3

For any $\widehat{f}(\mathbf{x}) = \widehat{\mathbf{w}}^\top \mathbf{x} + b_\star$, the coverage can be expressed as

$$\text{Coverage}(\widehat{f}) = \mathbb{P}\big(y \le \widehat{\mathbf{w}}^\top \mathbf{x} + b_\star\big) \stackrel{(i)}{=} \mathbb{P}\big(\mu_\star(\mathbf{x}) + \sigma_\star(\mathbf{x})z \le \widehat{\mathbf{w}}^\top \mathbf{x} + b_\star\big)$$

$$\stackrel{(ii)}{=} \mathbb{P}\big(\sigma_\star(\mathbf{x})(z - z_\alpha) \le (\widehat{\mathbf{w}} - \mathbf{w}_\star)^\top \mathbf{x}\big) = \mathbb{P}\left(z \le z_\alpha + \frac{(\widehat{\mathbf{w}} - \mathbf{w}_\star)^\top \mathbf{x}}{\sigma_\star(\mathbf{x})}\right)$$

$$= \mathbb{E}\left[\Phi_z\left(z_\alpha + \frac{(\widehat{\mathbf{w}} - \mathbf{w}_\star)^\top \mathbf{x}}{\sigma_\star(\mathbf{x})}\right)\right].$$

Above, (i) used the data distribution assumption (9), and (ii) follows by subtracting both sides by $\mu_\star(\mathbf{x}) + \sigma_\star(\mathbf{x}) = \mathbf{w}_\star^\top \mathbf{x} + b_\star$ by the linear true quantile assumption (10).

Now, by assumption $\alpha \ge 3/4$, we have $z_\alpha > z_{1/2} = 0$. We claim the following holds for all $t \in \mathbb{R}$:

$$\frac{1}{2}\big(\Phi_z(z_\alpha + t) + \Phi_z(z_\alpha - t)\big) \le \Phi_z(z_\alpha) - ct^2 \mathbf{1}\left\{|t| \le z_\alpha\right\}, \tag{52}$$

where $c > 0$ is a constant that only depends on $\Phi_z$ and $z_\alpha$. To see this, notice that $\Phi_z''(t) = \phi_z'(t) < 0$ for $t > 0$ and thus $\Phi_z$ is concave for $t \ge 0$. Further, $\Phi_z$ is $c$-strongly concave on $[z_\alpha/2, 3z_\alpha/2]$ for some $c > 0$ as $\Phi_z''(t) = \phi_z'(t)$ is continuous and negative on this compact interval. This shows that

$$\frac{1}{2}\big(\Phi_z(z_\alpha + t) + \Phi_z(z_\alpha - t)\big) \le \Phi_z(z_\alpha) - ct^2$$

for $|t| \le z_\alpha/2$, and further by the concavity of $\Phi_z$ on $[0, 2z_\alpha]$ that

$$\frac{1}{2}\big(\Phi_z(z_\alpha + t) + \Phi_z(z_\alpha - t)\big) \le \frac{1}{2}\big(\Phi_z(z_\alpha + t_0) + \Phi_z(z_\alpha - t_0)\big) \le \Phi_z(z_\alpha) - ct_0^2 \le \Phi_z(z_\alpha) - ct^2/4$$

for $|t| \in (z_\alpha/2, z_\alpha]$ (where $t_0 := z_\alpha/2$). This verifies claim (52) for $|t| \le z_\alpha$. On the other hand, if $|t| \ge z_\alpha$, we have (taking $t > 0$ w.l.o.g.) $\Phi_z(z_\alpha + t) \le 1$ always and $\overline{\Phi}(z_\alpha - t) \le \Phi_z(0) = 1/2$. Therefore

$$\frac{1}{2}\big(\Phi_z(z_\alpha + t) + \Phi_z(z_\alpha - t)\big) \le \frac{1}{2}(1 + 1/2) = 3/4 \le \Phi(z_\alpha).$$

This verifies claim (52) for $|t| > z_\alpha$.

Now, note that $(\widehat{\mathbf{w}} - \mathbf{w}_\star)^\top \mathbf{x}/\sigma_\star(\mathbf{x})$ is symmetric about $0$ by our assumption that $\mathbf{x}$ has a symmetric distribution and $\sigma_\star(\mathbf{x}) = \sigma_\star(-\mathbf{x})$. Therefore, we can rewrite and upper bound the coverage using (52):

$$\text{Coverage}(\widehat{f}) = \mathbb{E}\left[\frac{1}{2}\left(\Phi_z\left(z_\alpha + \frac{(\widehat{\mathbf{w}} - \mathbf{w}_\star)^\top \mathbf{x}}{\sigma_\star(\mathbf{x})}\right) + \Phi_z\left(z_\alpha - \frac{(\widehat{\mathbf{w}} - \mathbf{w}_\star)^\top \mathbf{x}}{\sigma_\star(\mathbf{x})}\right)\right)\right]$$

$$\stackrel{(i)}{\le} \mathbb{E}\left[\Phi_z(z_\alpha) - c\left(\frac{(\widehat{\mathbf{w}} - \mathbf{w}_\star)^\top \mathbf{x}}{\sigma_\star(\mathbf{x})}\right)^2 \mathbf{1}\left\{\left|\frac{(\widehat{\mathbf{w}} - \mathbf{w}_\star)^\top \mathbf{x}}{\sigma_\star(\mathbf{x})}\right| \le z_\alpha\right\}\right]$$

$$= \alpha - c\mathbb{E}\left[\left(\frac{(\widehat{\mathbf{w}} - \mathbf{w}_\star)^\top \mathbf{x}}{\sigma_\star(\mathbf{x})}\right)^2 \mathbf{1}\left\{\left|\frac{(\widehat{\mathbf{w}} - \mathbf{w}_\star)^\top \mathbf{x}}{\sigma_\star(\mathbf{x})}\right| \le z_\alpha\right\}\right]$$

$$\stackrel{(ii)}{\le} \alpha - \frac{c}{\overline{\sigma}^2}\mathbb{E}\left[\big((\widehat{\mathbf{w}} - \mathbf{w}_\star)^\top \mathbf{x}\big)^2 \mathbf{1}\left\{\big|(\widehat{\mathbf{w}} - \mathbf{w}_\star)^\top \mathbf{x}\big| \le z_\alpha \underline{\sigma}\right\}\right]$$

$$= \alpha - \frac{c}{\overline{\sigma}^2}\left((\widehat{\mathbf{w}} - \mathbf{w}_\star)^\top \mathbb{E}[\mathbf{x}\mathbf{x}^\top](\widehat{\mathbf{w}} - \mathbf{w}_\star) - \mathbb{E}\left[\big((\widehat{\mathbf{w}} - \mathbf{w}_\star)^\top \mathbf{x}\big)^2 \mathbf{1}\left\{\big|(\widehat{\mathbf{w}} - \mathbf{w}_\star)^\top \mathbf{x}\big| > z_\alpha \underline{\sigma}\right\}\right]\right)$$

$$\stackrel{(iii)}{\le} \alpha - \frac{c}{\overline{\sigma}^2}\left(\gamma \|\widehat{\mathbf{w}} - \mathbf{w}_\star\|_2^2 - \underbrace{\mathbb{E}\left[\big((\widehat{\mathbf{w}} - \mathbf{w}_\star)^\top \mathbf{x}\big)^2 \mathbf{1}\left\{\big|(\widehat{\mathbf{w}} - \mathbf{w}_\star)^\top \mathbf{x}\big| > z_\alpha \underline{\sigma}\right\}\right]}_{(\star)}\right).$$

Above, (i) used (52); (ii) used the bound $\underline{\sigma} \le \sigma_\star(\mathbf{x}) \le \overline{\sigma}$; (iii) used the covariance lower bound $\mathbb{E}[\mathbf{x}\mathbf{x}^\top] \succeq \gamma \mathbf{I}_d$. Further, letting $r := \|\widehat{\mathbf{w}} - \mathbf{w}_\star\|_2$, the random variable $(\widehat{\mathbf{w}} - \mathbf{w}_\star)^\top \mathbf{x}$ (with randomness only in $\mathbf{x}$) is $Kr^2$-sub-Gaussian, since $\mathbf{x}$ is $K$-sub-Gaussian by our assumption. Therefore the

term $(\star)$ can be further upper bounded as

$$
\begin{aligned}
(\star) &\le \left( \mathbb{E}\left[ \left( (\widehat{\mathbf{w}} - \mathbf{w}_\star)^\top \mathbf{x} \right)^4 \right] \cdot \mathbb{P}\left( |(\widehat{\mathbf{w}} - \mathbf{w}_\star)^\top \mathbf{x}| > z_\alpha \underline{\sigma} \right) \right)^{1/2} \\
&\le \left( C K^2 r^4 \cdot 2 \exp(-z_\alpha^2 \underline{\sigma}^2 / K r^2) \right)^{1/2} \\
&\le C K r^2 \cdot \exp(-z_\alpha^2 \underline{\sigma}^2 / 2 K r^2) \overset{(i)}{\le} \frac{1}{2} \underline{\gamma} r^2,
\end{aligned}
$$

where (i) happens if $r \le r_0$ for some $r_0 = r_0(\underline{\gamma}, \underline{\sigma}, K, z_\alpha)$. Plugging this back into the preceding bound yields

$$
\text{Coverage}(\widehat{f}) \le \alpha - \frac{c\underline{\gamma}}{2\overline{\sigma}^2} \cdot r^2 = \alpha - \frac{c\underline{\gamma}}{2\overline{\sigma}^2} \cdot \|\widehat{\mathbf{w}} - \mathbf{w}_\star\|_2^2
$$

for any $\widehat{\mathbf{w}}$ such that $\|\widehat{\mathbf{w}} - \mathbf{w}_\star\|_2 \le r_0$. This proves the desired result. $\qquad\square$

## F   Additional experimental details and ablations

### F.1   Simulations

We provide additional details about our simulations in Section 5.1. In each problem instance, we generate $(\mathbf{x}_i, y_i)$ from the Gaussian linear model (4): $\mathbf{x}_i \sim \mathsf{N}(\mathbf{0}, \mathbf{I}_d)$, $y_i = \mathbf{w}_\star^\top \mathbf{x}_i + z_i$ where $z_i \overset{\text{iid}}{\sim} P_z = \mathsf{N}(0, 0.25)$. We choose $\|\mathbf{w}_\star\|_2 = 1$. We run the (sub)-gradient descent algorithm on the full empirical risk $\widehat{R}_n$ (note the risk also depends on the quantile level $\alpha$) for 50k steps, with initial learning rate 0.01 and a 10x learning rate decay at the 25k-th step. For all our settings (choice of $n, d, \alpha$), this optimization schedule ensures that the training loss changes by less than $10^{-5}$ between consecutive iterations at the final iteration.

Each problem instance yields a solution $(\widehat{\mathbf{w}}, \widehat{b})$ which specifies a linear quantile function $\widehat{f}(\mathbf{x}) = \widehat{\mathbf{w}}^\top \mathbf{x} + \widehat{b}$. We evaluate its coverage *exactly* using the closed-form formula (cf. Section 6)

$$
\text{Coverage}(\widehat{f}) = \mathbb{E}_{G \sim \mathsf{N}(0,1)}\left[ \Phi_z\left( \|\widehat{\mathbf{w}} - \mathbf{w}_\star\|_2\, G + \widehat{b} \right) \right].
$$

We compute this by using numerical integration (over the gaussian random variable $G$). The entire set of experiments (for producing Figure 1) is done on a single CPU machine in roughly 6 hours.

### F.2   Real data experiments

We provide additional details about our real data experiments in Section 5.2 and 5.3. All models (linear, MLP, MLP-freeze) in Section 5.2 are trained by minimizing the quantile risk (3). We use SGD with momentum 0.9, initial learning rate $10^{-3}$ for 1500 epochs, and apply a 10x learning rate decay at epoch $\{500, 1000\}$. For each dataset and each random seed, we perform a train-validation split where we use $80\%$ of the data as the train set and $20\%$ of the data as the test set. The coverage of the trained model is evaluated on the test split. For all datasets and all models, we repeat the same experiment across 8 random seeds, and report the mean and standard deviation of the coverage in Table 1.

For our pseudo-label experiments in Section 5.3, we train the linear model $\widehat{\mathbf{w}}$ first by minimizing the square loss and using the same optimization schedule above. After $\widehat{\mathbf{w}}$ is learned, we generate the pseudo-labels $y_i^{\text{pseudo}}$ using $\widehat{\mathbf{w}}$ and the estimated standard deviation $\widehat{\sigma}$ as described in Section 5.3. This is done for both the train and test sets for which we obtain a "pseudo" train set and a "pseudo" test set. We then perform linear quantile regression on these pseudo datasets in a same fashion as in Section 5.3.

The experiments for Sections 5.2 and 5.3 are done on a 8-GPU machine (with Tesla V-100 GPUs) in roughly a day.

**Ablations on $\alpha$**   Table 3 and 4 report coverage results on the real data with $\alpha \in \{0.8, 0.95\}$ respectively, in the same settings as in Section 5.2. These tables also show that under-coverage happens consistently across different datasets and different models, with patterns similar as in Table 1 (which uses $\alpha = 0.9$).

**Table 3:** Coverage (%) of quantile regression on real data at nominal level $\alpha = 0.8$. Each entry reports the test-set coverage with mean and std over 8 random seeds. $(d, n)$ denotes the {feature dim, # training examples}.

| Dataset | Linear | MLP-3-64 | MLP-3-512 | MLP-freeze-3-512 | $d$ | $n$ |
|---|---|---|---|---|---|---|
| Community | 78.25±1.75 | 66.07±1.48 | 56.17±2.81 | 77.45±1.76 | 100 | 1599 |
| Bike | 79.95±0.66 | 78.07±1.00 | 78.66±0.86 | 79.46±0.83 | 18 | 8708 |
| Star | 79.97±2.37 | 72.95±1.83 | 59.26±1.41 | 78.42±2.04 | 39 | 1728 |
| MEPS_19 | 80.11±1.12 | 76.47±0.93 | 70.04±0.75 | 79.02±1.28 | 139 | 12628 |
| MEPS_20 | 79.84±0.75 | 77.11±0.73 | 71.88±0.87 | 79.29±0.53 | 139 | 14032 |
| MEPS_21 | 79.57±0.72 | 74.58±0.70 | 65.55±0.69 | 79.29±0.73 | 139 | 12524 |
| Nominal ($\alpha$) | 80.00 | 80.00 | 80.00 | 80.00 | - | - |

**Table 4:** Coverage (%) of quantile regression on real data at nominal level $\alpha = 0.95$. Each entry reports the test-set coverage with mean and std over 8 random seeds. $(d, n)$ denotes the {feature dim, # training examples}.

| Dataset | Linear | MLP-3-64 | MLP-3-512 | MLP-freeze-3-512 | $d$ | $n$ |
|---|---|---|---|---|---|---|
| Community | 93.82±0.98 | 86.23±1.43 | 74.38±1.86 | 93.58±1.33 | 100 | 1599 |
| Bike | 94.56±0.45 | 93.77±0.63 | 93.16±0.80 | 94.19±0.65 | 18 | 8708 |
| Star | 94.08±1.73 | 90.96±1.91 | 81.58±1.82 | 93.39±1.68 | 39 | 1728 |
| MEPS_19 | 94.69±0.41 | 90.71±0.72 | 85.32±1.23 | 94.19±0.42 | 139 | 12628 |
| MEPS_20 | 94.84±0.30 | 92.06±0.43 | 87.32±0.77 | 94.58±0.32 | 139 | 14032 |
| MEPS_21 | 94.97±0.34 | 89.55±0.39 | 80.70±0.79 | 94.42±0.29 | 139 | 12524 |
| Nominal ($\alpha$) | 95.00 | 95.00 | 95.00 | 95.00 | - | - |

### F.3  License of datasets

The Community [2] and Bike [1] datasets are retrieved from the publicly available UCI machine learning repository [22] and subject to the license of the repository. The STAR dataset [6] is also a public access dataset. The three mediecal expenditure survey datasets MEPS_19, MEPS_20, MEPS_21 contain a data use agreement section in their documentation (cf. the "documentation" link in [3–5]) which our use case (train quantile functions and report coverages) comply with. All the datasets are anonymized and to the best of our knowledge do not contain personally identifiable information or offensive contents.