# OpenReview forum: "Understanding the Under-Coverage Bias in Uncertainty Estimation"
_NeurIPS.cc/2021/Conference — NeurIPS 2021 Spotlight_

### Official Review · Reviewer_FCrS · 2021-07-14

**Rating:** 6
**Confidence:** 4

**Summary:**

In this work, the authors consider linear quantile regression, paying particular attention to the "coverage" properties of the resulting predictor. The traditional goal of quantile regression is to learn a quantile of the conditional distribution of output $Y$, conditioned on the inputs $X$. Denoting the predictor returned by any quantile regression algorithm by $\hat{f}(\cdot)$, one can use this to construct a predictive interval for a freshly-drawn $Y$. The authors consider the one-sided interval $C(X) = (-\infty, \hat{f}(X)]$, and the event $\\{ Y \in C(X) \\}$. The authors call the probability of this event (over the draw of a pair $(X,Y)$) the coverage of $\hat{f}$. Their main results elucidate conditions under which the traditional pinball-loss based ERM quantile regression solution is biased below the desired quantile level, that is, the predictive interval tends to be "too small," due to $\hat{f}$ under-estimating the desired quantile.

In particular, they show that in the limit where the sample size and dimensionality are proportional, the coverage (the aforementioned probability, which depends on $\hat{f}$ and thus the training data) converges below the desired quantile, with this error depending directly on the ratio of dimension to sample size $d/n$. Furthermore, they show (under fixed $n$ and $d$) how for a certain family of data distributions and a linear quantile model, given any estimator of the weights, the downward bias term scales with the weight estimation error. Their empirical tests also effectively highlight the phenomena described formally.

**Limitations And Societal Impact:**

Please see the main review above.

**Main Review:**

Overall, the paper is quite clearly written, easy to follow, and the authors describe their contributions in a transparent fashion. My only reservations are related to the significance of the results, particularly for the machine learning community. As the authors mention, there are plenty of ways of getting around the aforementioned under-estimation bias, and in the context of "uncertainty estimation," it is arguably more common to use quantile regression as a sub-routine in a larger conformal prediction type of procedure,  rather than to pay particular attention to the coverage of the quantile regressor itself. The results here are of a fundamentally statistical nature, which I still think is fine for NeurIPS, but considering that quantile regression is a now-classical and well-studied topic, I would be surprised if similar insights did not already exist in the statistics literature, but I am not sufficiently familiar with that literature. Taking these points together, my recommendation is a borderline accept.

**Update:**
I thank the authors for their response. Having read through the authors' response and the other reviews, my overall opinion remains unchanged.

**Time Spent Reviewing:**

1

---

> ### Author Response · Authors · 2021-08-09
> **Response**
>
> Thank you for your thoughtful reviews!
>
> **“... significance of the results, particularly for the machine learning community… arguably more common to use quantile regression as a sub-routine in a larger conformal prediction type of procedure, rather than to pay particular attention to the coverage of the quantile regressor itself”**
>
> We agree that there are more advanced algorithms that build on quantile regression as a subroutine (e.g. the conformal quantile regression of Romano et al. 2019). However, we believe understanding the behavior of the quantile regressor itself is also worthy, as it is a classical and arguably quite simple algorithm, achieves consistency with infinite samples, but is observed to often under-cover in reality when used alone. As quantile regression is still used a lot, our theoretical results could offer helpful insights to when / how much it tends to under-cover, even when it is used as a subroutine of a larger procedure.
>
> **“... considering that quantile regression is a now-classical and well-studied topic… would be surprised if similar insights did not already exist in the statistics literature”**
>
> We believe the main reason why such results did not exist in the literature is because this under-coverage bias shows up when $d,n$ are large and on the same scale, and in theory this can only be captured by the proportional limit regime where $n,d\to\infty$ and $d/n=O(1)$. Existing theories either consider the asymptotic case ($d$ fixed, $n\to\infty$), in which case the coverage converges correctly to $\alpha$, or provide finite-sample error bounds in finite $n,d$ case but do not tell the sign of the coverage error (cf. Line 181 - 190 in the paper for more discussions). Technical tools for analyzing ERM problems in the proportional limit regime only appeared in the past few years (e.g. Donoho and Montanari 2016; Thrampoulidis et al. 2018), which may explain why previous work on quantile regression has not considered this regime and reached similar conclusions.

---

### Official Review · Reviewer_bjfV · 2021-07-16

**Rating:** 8
**Confidence:** 3

**Summary:**

This paper gives theoretical insights into the under-coverage phenomenon often observed in quantile regression, i.e., for a continuous distribution, the probability of observing  a true label below a learned  $\alpha$-quantile is often below $\alpha$, which is consider a failure in estimating the uncertainty.
Two important results are shown. The first one provides a characterization of the under-coverage in the setting of a Gaussian linear model in the under-parameterized high-dimensional proportional limit (i.e. $d,n \rightarrow \infty$ , $d/n\rightarrow\kappa$ with small $\kappa$) independently of the noise distribution.
The experimental results show that the formulas match pretty well the observed coverage in realistic situations of dimension around 100, on synthetic and real data.
The second important result shows that the source of the under-coverage stems essentially from the error in estimating the high-dimensional linear coefficient, under some assumptions on the underlying source.

**Limitations And Societal Impact:**

Limitations are clearly related to the mathematical assumptions and are properly discussed.  They are reasonable to show this kind of results and do not make them less relevant or interesting for future research.

The authors do not foresee any negative societal impact of this work because
it is a theoretical paper on understanding existing uncertainty estimation algorithms.



**Main Review:**

**Originality**: As far as I know, the analysis is new and sheds light on a known but unexplained phenomenon. It builds on recent advances in high-dimensional  M-estimation but introduces a novel concentration argument to deal with the additional learnable bias, which was not considered in previous works.

**Quality**: This paper is a complete and sound piece of work addressing an important problem of quantile regression.I am not familiar with all the technical elements of the proofs in the Appendix, so I can't say too much about their correctness.  The proof overviews in the main text are helpful.  The noise smoothness assumptions of the main theorem seem reasonable and those of theorem 3 allow to consider more general data distributions. The experimental results nicely match the proposed theory in realistic situations.

**Clarity**: The paper is very well organized and written which makes it very pleasant to read. There is a very  good literature review that positions it with respect to prior work.

**Significance**: these results will be of interest for the machine learning communities working on quantile regression to potentially build on these results and extend them to more general settings.


**Minor comments:**

Assumption A should specify what is $\alpha$ , is it $\alpha\in(0.5,1)$  ?

Line 162:  "small" should be more precise

Line 192: I think it should say "under-coverage" instead of "over-coverage"

Line 245: K-sub-Gaussian should be defined

Line 252: "small" should be more precise

References:
arXiv versions should be replaced by their peer-reviewed versions when possible, e.g.:

Y. Romano, E. Patterson, and E. J. Candès. Conformalized quantile regression. arXiv preprint
471 arXiv:1905.03222, 2019.

[57] is incomplete


**Time Spent Reviewing:**

5

---

> ### Author Response · Authors · 2021-08-09
> **Response**
>
> Thank you for your positive feedback and the thoughtful comments!
>
> **Assumption A on $\alpha$**
>
> Yes, it should be a fixed $\alpha\in(0.5, 1)$. We specified this on Line 118 (Section 2). We will also restate this in Assumption A (to further clarify) in our revision.
>
> **Comments on typos and references**
>
> Thank you for spotting these! We will fix them in our revision.

---

### Official Review · Reviewer_SQgW · 2021-07-16

**Rating:** 7
**Confidence:** 3

**Summary:**

Quantifying predictive uncertainty is a crucial task for high-stakes prediction problems. The paper focus on quantile regression, a common approach for quantifying predictive uncertainty in regression problems. Authors consider realizable linear setting when the true model belongs to the considered functional class. With a focus on the under-parameterized setting, they theoretically justify the under-coverage of the learnt quantile regressors. Established theoretical results are backed up with supporting empirical evidence.

**Limitations And Societal Impact:**

The authors mention the limitations of a simplified setting considered in the work. Still, it would be nice to have a more detailed discussion section (with more focus on the impact of the established results).

**Main Review:**

The paper is well-written making it easy to follow for a non-expert in the area. The authors discuss the ways in which the results presented in the paper differ from the ones available in the literature. The main contribution of the paper is on the theoretical side, and it seems quite interesting and convincing. While pure reliance on prediction intervals constructed through learning two separate quantiles could be unfounded (which particularly motivated applying conformal techniques for performing appropriate corrections), studying the nature of the underlying biases, even in simplified settings, is important.

Couple of minor typos:
--- line 98 --> with quantile regression to correct ITS coverage bias
--- line 192  --> under-coverage?
--- line 215 --> $w_\star^\top x + z_\alpha$ should be present instead of $w_\star^\top + z_\alpha$.

**Time Spent Reviewing:**

4

---

> ### Author Response · Authors · 2021-08-09
> **Response**
>
> Thank you for your positive feedback and comments on our paper!
>
> **Typos; Detailed discussion on the impact and limitations**
>
> Thanks for the suggestions, and efforts in finding the typos. We will add some more detailed discussions regarding our setting and the impact of our results in our revision. We will also fix our typos.

---

### Official Review · Reviewer_sjsB · 2021-07-18

**Rating:** 6
**Confidence:** 3

**Summary:**

This paper presents a theoretical study on the coverage of quantile regression algorithms. More specifically, the authors prove theoretically that linear quantile regression exhibits an inherent under-coverage bias, with an order d/n difference in the marginal coverage regardless of the noise distribution (but Gaussian for the covariates). They also show that the primary source of this under-coverage bias is the estimation error in the linear coefficients, while the estimation error in the bias term can have either an under- or over-coverage effect. Numerical experiments further support their findings.

**Limitations And Societal Impact:**

The authors only study linear quantile regression under specific assumptions, and the theoretical results are viewed as somewhat restrictive. On the other hand, it sets up a good direction and opens up new questions to explore. This paper is mostly a theoretical work and I do not foresee any negative societal impact.

**Main Review:**

Quality and Significance:

Pros: The authors give a clear contribution in explaining the under-coverage phenomenon in linear quantile regression. The analysis, as the authors claim, is unlike existing results in that it captures the sign of the bias, and is the first rigorous theoretical justification on the under-coverage phenomenon. The study can open up more research questions along this line.

Cons: The biggest limitation seems to be that only the classical linear quantile regression algorithm with pinball loss is analyzed, and it is unclear whether the insight is generalizable. As the authors are aware, there are many advanced uncertainty estimation algorithms (e.g., conformal-prediction-based approaches, quantile regression forests etc.), and the paper would be much stronger if similar insights could be drawn to these advanced approaches. For instance, the bias term which can have either under- or over-coverage effect seems to be very specific to the linear model, and it is unclear any generalization of such insights could be drawn to other models.

Related to above, a relatively milder concern is that Theorem 1 requires that both n and d go to infinity. I don't know how easy/meaningful it is to "model" an increasing data dimension beyond linear (although I understand the authors' motivation in letting d go to infinity to highlight the nontrivial impact of the under-coverage bias).


Clarity:

The paper is well-written. It is easy to read and understand the main theorems, and intuitions are well presented.

In line 192, “the over-coverage should get more severe as κ—the measure of over-parametrization in this problem—gets larger”. Should “over-coverage” be under-coverage bias?


Originality:

The authors present a rigorous theoretical study on the under-coverage of uncertainty estimation algorithms in learning quantiles, focusing on linear quantile regression. The analysis giving rise to the sign of the coverage bias seems novel.

On the other hand, except for the sign of the coverage bias, the finite-sample estimation error of linear quantile regression is well-studied, e.g., in Steinwart et al., 2011.

**Time Spent Reviewing:**

3 hours

---

> ### Author Response · Authors · 2021-08-09
> **Response**
>
> Thank you for your thoughtful comments on our paper!
>
> **“only the classical linear quantile regression algorithm with pinball loss is analyzed... many advanced uncertainty quantification algorithms… paper would be much stronger if similar insights could be drawn to these advanced approaches…”**
>
> We first emphasize that the main focus of this paper is analyzing the exact coverage error including its sign and magnitude, and this was not understood even in the linear case (and cannot be deduced from existing theories on quantile regression). Therefore, we began with the linear model to have a precise understanding, which we found already requires new analyses. To complement this, in our experiments (Section 5.2) we test quantile regression with pinball loss + neural networks, and show that the under-coverage still happens (and is even more severe than the linear quantile functions). That said, we agree that theoretically analyzing more advanced algorithms such as quantile forests or conformal-based approaches would be an interesting direction for future work.
>
> **“Theorem 1 requires both $n$ and $d$ go to infinity… don’t know how easy / meaningful it is to ‘model’ an increasing data dimension beyond linear”**
>
> By increasing the dimension $d$, we meant for each $d$ we assume a different data distribution (in our case Gaussian linear model in $d$ dimensions), look at the behavior of the algorithm on this distribution, then let $d$ go to infinity together with the sample size $n$. Such limits can be properly defined for non-linear models too, although in prior work it is indeed the case that this limit is usually considered on linear models. We will clarify this in our revision.
>
> We also remark that although taking the limit is required in theory, in simulations we observed that $d=100$, $n=200$ is already enough for the under-coverage bias to closely match our theoretical predictions at $d/n=\kappa=0.5$ (Figure 1, Section 5.1). Therefore, from a more practical point of view, our conclusions hold at rather realistic values of $n,d$ and do not require them to be excessively large.
>
> **“Except for the sign of the coverage bias, finite-sample estimation error of linear quantile regression is well-studied, e.g. in Steinwart et al. 2011”**
>
> As we answered above, the sign of the coverage bias is indeed our motivating question, which we found was not well-studied in prior works (such as the finite-sample analysis of Steinwart et al. 2011). We also remark that our Theorem 1 not only tells the sign, but also shows the precise limiting value of the under-coverage bias in Gaussian linear models to be $C_{\alpha, \kappa}=\alpha - (\alpha-1/2)\kappa + o(\kappa)$ when $\kappa=d/n$ is small. This magnitude is also not present in the finite-sample analysis of Steinwart et al. (2011) as it only gives an upper bound on the coverage error.
>
> **Line 192 “over-coverage -> under-coverage”**
>
> Thanks for spotting this, it was our typo indeed. We will fix this in our revision.

---

### Decision · Program_Chairs · 2021-09-27

**Decision:**

Accept (Spotlight)

**Comment:**

This article provides a theoretical explanation for the empirical observation that quantile regression tends to exhibit an under-coverage bias, in the sense of frequentist coverage of prediction intervals.  The theory shows that this under-coverage stems from the error in estimating high-dimensional parameters when the number of parameters $d$ grows proportionally with the sample size $n$.  An explicit formula for the bias is provided (Eqn 7) in the case of a linear Gaussian model when $d/n$ is small, under quite general conditions on the noise distribution.  This under-coverage occurs even in the well-specified setting of learning a linear quantile function when the true data distribution follows a linear Gaussian model.  The theory is validated in empirical experiments on simulated and real data.

The paper is well-written and successfully addresses an important problem in a compelling way.  To my knowledge, the results are novel.  The reviews were consistently positive, and I believe the paper provides a valuable contribution to the literature.  The experimental results on neural networks (e.g., in Table 1) are particularly interesting and relevant to the NeurIPS community.

The main limitation (as mentioned by Reviewer sjsB) is that the analysis considers only the classical linear quantile regression algorithm with pinball loss.  It would be interesting to extend the results to more advanced algorithms.  However, this is a minor limitation and the paper opens up directions of future research into such extensions.